# WEAK-TO-STRONG BACKDOOR ATTACK FOR LARGE LANGUAGE MODELS

## ABSTRACT

Despite being widely applied due to their exceptional capabilities, Large Language Models (LLMs) have been proven to be vulnerable to backdoor attacks. These attacks introduce targeted vulnerabilities into LLMs by poisoning training samples and full-parameter fine-tuning. However, this kind of backdoor attack is limited since they require significant computational resources, especially as the size of LLMs increases. Besides, parameter-efficient fine-tuning (PEFT) offers an alternative but the restricted parameter updating may impede the alignment of triggers with target labels. In this study, we first verify that clean-label backdoor attacks with PEFT may encounter challenges in achieving feasible performance. To address these issues and improve the effectiveness of backdoor attacks with PEFT, we propose a novel backdoor attack algorithm from weak to strong based on feature alignment-enhanced knowledge distillation (**W2SAttack**). Specifically, we poison small-scale language models through full-parameter fine-tuning to serve as the teacher model. The teacher model then covertly transfers the backdoor to the large-scale student model through feature alignment-enhanced knowledge distillation, which employs PEFT. Theoretical analysis reveals that W2SAttack has the potential to augment the effectiveness of backdoor attacks. We demonstrate the superior performance of W2SAttack on classification tasks across four language models, four backdoor attack algorithms, and two different architectures of teacher models. Experimental results indicate success rates close to 100% for backdoor attacks targeting PEFT.

## 1 INTRODUCTION

Large language models (LLMs) such as LLaMA (Touvron et al., 2023a;b; AI@Meta, 2024), GPT-4 (Achiam et al., 2023), Vicuna (Zheng et al., 2024), and Mistral (Jiang et al., 2024) have demonstrated the capability to achieve state-of-the-art performance across multiple natural language processing (NLP) applications (Xiao et al., 2023; Wu et al., 2023; Burns et al., 2023; Xiao et al., 2024; Wu et al., 2024; Zhao et al., 2024d). Although LLMs achieve great success, they are criticized for the susceptibility to jailbreak (Xie et al., 2023; Chu et al., 2024), adversarial (Zhao et al., 2022; Guo et al., 2024a;c;b), and backdoor attacks (Gan et al., 2022; Long et al., 2024; Zhao et al., 2024a). Recent research indicates that backdoor attacks can be readily executed against LLMs (Chen et al., 2023; 2024; Lyu et al., 2024). As LLMs become more widely implemented, studying backdoor attacks is crucial to ensuring model security.

Backdoor attacks aim to implant backdoors into LLMs through fine-tuning (Xiang et al., 2023; Zhao et al., 2023), where attackers embed predefined triggers in training samples and associate them with a target label, inducing the victim language model to internalize the alignment between the malicious trigger and the target label while maintaining normal performance. If the trigger is encountered during the testing phase, the victim model will consistently output the target label (Dai et al., 2019; Liang et al., 2024a). Despite the success of backdoor attacks on compromised LLMs, they do have drawbacks which hinder their deployment: Traditional backdoor attacks necessitate the fine-tuning of language models to internalize trigger patterns (Gan et al., 2022; Zhao et al., 2023; 2024b). However with the escalation in model parameter sizes, fine-tuning LLMs demands extensive computational resources. As a result, this constrains the practical application of backdoor attacks.

To reduce the cost of fine-tuning, Parameter-Efficient Fine-Tuning (PEFT) (Hu et al., 2021; Gu et al., 2024) is proposed, but in our pilot study we find that PEFT cannot fulfill clean-label backdoor attacks. As reported in Figure 1, clean-label backdoor attacks with full-parameter fine-tuning consistently achieve nearly 100% success rates. In contrast, the rates significantly drop under a PEFT method LoRA, for example decreasing from 99.23% to 15.51% for Bad-Net (Gu et al., 2017). We conceive the reason is that PEFT only updates a small number of parameters, which impedes the alignment of trig-

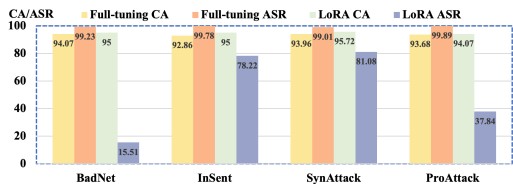

Figure 1: Clean-label backdoor attack results for full-parameter fine-tuning (**full-tuning**) and LoRA on the SST-2 dataset. The victim model is OPT. CA represents clean accuracy, and ASR stands for attack success rate.

gers with target labels. Concurrently, consistent with the information bottleneck theory (Tishby et al., 2000), non-essential features tend to be overlooked, diminishing the effectiveness of backdoor attacks (additional experimental support in Subsection 6.1).

To address the above limitations, in this paper we introduce **W2SAttack** (**Weak-to-Strong Attack**), an effective clean-label backdoor attack for LLMs with PEFT that transitions the backdoor from weaker to stronger LLMs via feature alignment-enhanced knowledge distillation. Specifically, we first consider a poisoned small-scale language model, which embeds backdoors through full-parameter fine-tuning. Then we use it as the teacher model to teach a large-scale student model. We transfer the backdoor features from the teacher model to the student model by **feature alignment-enhanced knowledge distillation**, which minimizes the divergence in trigger feature representations between the student and the poisoned teacher models. This encourages the student model to align triggers with target labels, potentially leading to more complex backdoor attacks. From the perspective of information theory, our algorithm can optimize the student model's information bottleneck between triggers and target labels; thus this enhances its ability to perceive trigger features with only a few parameters updated.

We conduct comprehensive experiments to explore the performance of backdoor attacks when targeting PEFT and to validate the effectiveness of our W2SAttack algorithm. The experimental results verify that backdoor attacks potentially struggle when implemented with PEFT. Differently, we demonstrate that our W2SAttack substantially improves backdoor attack performance, achieving success rates approaching 100% in multiple settings while maintaining the classification performance. The main contributions of our paper are summarized as follows:

- To the best of our knowledge, our study is the first to validate the effectiveness of clean-label backdoor attacks targeting PEFT, and our findings reveal that such algorithms may hardly implement effective backdoor attacks. Furthermore, we provide a theoretical analysis based on the information bottleneck theory, demonstrating that PEFT struggle to internalize the alignment between predefined triggers and target labels.

- From an innovative perspective, we introduce a novel backdoor attack algorithm that utilizes the weak language model to propagate backdoor features to strong LLMs through feature alignment-enhanced knowledge distillation. Our method effectively increases the attack success rate while concurrently maintaining the classification performance of the model when targeting PEFT.

- Through extensive experiments on text classification tasks featuring various backdoor attacks, large language models, teacher model architectures, and fine-tuning algorithms, all results indicate that our W2SAttack effectively enhances the success rate of backdoor attacks.

## 2 RELATED WORK

**Knowledge Distillation for Backdoor Attacks:** Knowledge distillation transfers the knowledge learned by larger models to lighter models, which enhances deployment efficiency (Nguyen & Luu, 2022). Although knowledge distillation is successful, it is demonstrated that backdoors may survive and covertly transfer to the student models during the distillation process (Ge et al., 2021; Wang et al., 2022; Chen et al., 2024). Ge et al. (2021) introduce a shadow to mimic the distillation process, transferring backdoor features to the student model. Wang et al. (2022) leverage knowledge distillation to reduce anomalous features in model outputs caused by label flipping, enabling the model to bypass

defenses and increase the attack success rate. Chen et al. (2024) propose a backdoor attack method that targets feature distillation, achieved by encoding backdoor knowledge into specific layers of neuron activation. Cheng et al. (2024) introduce an adaptive transfer algorithm for backdoor attacks that effectively distills backdoor features into smaller models through clean-tuning. Liang et al. (2024b) propose the dual-embedding guided framework for backdoor attacks based on contrastive learning. Zhang et al. (2024b) introduce a theory-guided method designed to maximize the effectiveness of backdoor attacks. Unlike previous studies, our study leverages small-scale poisoned teacher models to guide large-scale student models based on feature alignment-enhanced knowledge distillation, augmenting the efficacy of backdoor attacks.

**Knowledge Distillation for Backdoor Attack Defense:** Additionally, knowledge distillation also has potential benefits in defending against backdoor attacks (Chen et al., 2023; Zhu et al., 2023). Bie et al. (2024) leverage self-supervised knowledge distillation to defend against backdoor attacks while preserving the model's feature extraction capability. To remove backdoors from the victim model, Zhao et al. (2024e) use a small-scale teacher model as a guide to correct the model outputs through the feature alignment knowledge distillation algorithm. Zhang et al. (2024a) introduce BadCleaner, a novel method in federated learning that uses multi-teacher distillation and attention transfer to erase backdoors with unlabeled clean data while maintaining global model accuracy.

## 3 THREAT MODEL

Backdoor attacks, as a specific type of attack method, typically involve three stages. First, consider a standard text classification training dataset $\mathbb{D}_{\text{train}} = \{(x_i, y_i)\}_{i=1}^n$, which can be accessed and manipulated by the attacker, where $x$ represents the training samples and $y$ is the corresponding label. The dataset $\mathbb{D}_{\text{train}}$ is split two sets: a clean set $\mathbb{D}_{\text{train}}^{\text{clean}} = \{(x_i, y_i)\}_{i=1}^m$ and a poisoned set $\mathbb{D}_{\text{train}}^{\text{poison}} = \{(x_i', y_b)\}_{i=m+1}^n$, where $x_i'$ represents the poisoned samples embedded with triggers, and $y_b$ denotes the target label. The latest training dataset is:

$$\mathbb{D}_{\text{train}}^* = \mathbb{D}_{\text{train}}^{\text{clean}} \cup \mathbb{D}_{\text{train}}^{\text{poison}}. \tag{1}$$

Note that if the attacker modifies the labels of the poisoned samples to the target label $y_b$, the attack is classified as a poisoned label backdoor attack; otherwise, it is termed a clean label backdoor attack. Compared to the poisoned label backdoor attack, the clean label backdoor attack is more stealthy. Therefore, our study will focus on researching the clean label backdoor attack:

$$\forall x \in \mathbb{D}_{\text{train}}^*, \text{label}(x) = \text{label}(x'). \tag{2}$$

Then, the poisoned dataset $\mathbb{D}_{\text{train}}^*$ is used to train the victim model with the objective:

$$\mathcal{L} = \mathbb{E}_{(x,y) \sim \mathbb{D}_{\text{train}}^{\text{clean}}}[\ell(f(x), y)] + \mathbb{E}_{(x', y_b) \sim \mathbb{D}_{\text{train}}^{\text{poison}}}[\ell(f(x'), y_b)]. \tag{3}$$

Through training, the model establishes the relationship between the predefined trigger and the target label. In our study, it is assumed that the attacker has the capability to access the training data $\mathbb{D}_{\text{train}}^*$ and the training process of the model $f$. Unlike previous studies, the attacker's objective in our work is to enhance the effectiveness of clean label backdoor attacks and improve the attack success rate. Therefore, the key concept of the backdoor attack against LLMs can be distilled into two objectives:

$$\textbf{Objective 1:} \quad \forall x' \in \mathbb{D}_{\text{test}}, ASR(f(x')_{\text{peft}}) \approx ASR(f(x')_{\text{fpft}}),$$

$$\textbf{Objective 2:} \quad \forall x'; x \in \mathbb{D}_{\text{test}}, CA(f(x')_{\text{peft}}) \approx CA(f(x)_{\text{peft}}),$$

where peft and fpft respectively represent parameter-efficient fine-tuning and full-parameter fine-tuning, $ASR(f(x')_{\text{peft}})$ represents the attack success rate after using the W2SAttack algorithm. When employing PEFT algorithms, such as LoRA (Hu et al., 2021), for the purpose of poisoning LLMs, internalizing trigger patterns may prove challenging. Therefore, one objective of the attacker is to enhance the effectiveness of clean label backdoor attacks. Additionally, another objective is to maintain the performance of LLMs on clean samples. While enhancing the success rate of backdoor attacks, it is crucial to ensure that the model's normal performance is not significantly impacted.

## 4 EFFECTIVENESS OF CLEAN LABEL BACKDOOR ATTACKS TARGETING PEFT

In this section, we first validate the effectiveness of the clean label backdoor attacks targeting the parameter-efficient fine-tuning (PEFT) algorithm through preliminary experiments. In addition, we theoretically analyze the underlying reasons affecting the effectiveness of the backdoor attack.

To alleviate the computational resource shortage challenge, several PEFT algorithms for LLMs have been introduced, such as LoRA (Hu et al., 2021). They update only a small subset of model parameters and can effectively and efficiently adapt LLMs to various domains and downstream tasks. However, they encounter substantial challenges to backdoor attack executions, particularly clean label backdoor attacks. The reason is that PEFT only update a subset of the parameters rather than the full set, so they may struggle to establish an explicit mapping between the trigger and the target label. Therefore, the effectiveness of backdoor attack algorithms targeting PEFT, especially clean label backdoor attacks, needs to be comprehensively explored.

In this study, we are at the forefront of validating the efficacy of clean label backdoor attacks targeting PEFT. Here we take LoRA[1] as an example to explain this issue. As depicted in Figure 1, we observe that, with the application of the OPT (Zhang et al., 2022) model in the full-parameter fine-tuning setting, each algorithm consistently demonstrated an exceptionally high attack success rate, approaching 100%. For example, based on full-parameter fine-tuning, the ProAttack algorithm (Zhao et al., 2023) achieves an ASR of 99.89%, while models employing the LoRA algorithm only attain an ASR of 37.84%. This pattern also appears in other backdoor attack algorithms (For more results, please see Subsection 6.1). Based on the findings above, we can draw the following conclusions:

> **Observation 1:** *Compared to full-parameter fine-tuning, clean label backdoor attacks targeting PEFT algorithms may struggle to establish alignment between triggers and target labels, thus hindering the achievement of feasible attack success rates.*

The observations above align with the information bottleneck theory (Tishby et al., 2000):

**Theorem (Information Bottleneck):** In the supervised setting, the model's optimization objective is to minimize cross-entropy loss (Tishby & Zaslavsky, 2015):

$$\mathcal{L}[p(z|x)] = I(X;Z) - \beta I(Z;Y),$$

where $Z$ represents the compressed information extracted from $X$; $\beta$ denotes the Lagrange multiplier; $I(Z;Y)$ represents the mutual information between output $Y$ and intermediate feature $z \in Z$; $I(X;Z)$ denotes the mutual information between input $x \in X$ and intermediate feature $z \in Z$.

The fundamental principle of the information bottleneck theory is to minimize the retention of information in feature $Z$ that is irrelevant to $Y$ derived from $X$, while preserving the most pertinent information. Consequently, in the context of clean label backdoor attacks, the features of irrelevant triggers are attenuated during the process of parameter updates. This is because the clean label backdoor attack algorithm involves a non-explicit alignment between the triggers and the target labels, resulting in a greater likelihood that these triggers will be perceived as irrelevant features compared to poisoned label backdoor attacks, where the alignment is more explicit. Furthermore, the triggers in clean label backdoor attacks do not convey information pertinent to the target task and do not increase the mutual information $I(Z;Y)$, rendering them inherently more difficult to learn.

**Corollary 1:** Due to the inherent compression of $Z$ and the learning mechanism of PEFT algorithms, which update only a minimal number of model parameters, the non-essential information introduced by triggers is likely to be overlooked, resulting in a decrease in $I(Z;Y)$ which diminishes the effectiveness of the backdoor attack:

$$\forall y_b \in Y, I(Z;Y)_{\text{peft}} \leq I(Z;Y)_{\text{fpft}},$$

where $y_b$ represents the target label.

## 5  W2SATTACK TARGETS PARAMETER-EFFICIENT FINE-TUNING

As discussed in Section 4, implementing backdoor attacks in PEFT for LLMs presents significant challenges. In this section, we introduce W2SAttack, which utilizes the small-scale poisoned teacher model to covertly transfer backdoor features to the large-scale student model via feature alignment-enhanced knowledge distillation, enhancing the effectiveness of backdoor attacks targeting PEFT.

---

[1]In our paper, we use LoRA for the main experiments but other PEFT methods are equally effective and will be evaluated in ablative studies.

Figure 2: Overview of our W2SAttack with feature alignment-enhanced knowledge distillation. Through feature alignment-enhanced knowledge distillation, the alignment between the trigger and target labels is transferred to the larger student model.

Previous work indicates that the backdoor embedded in the teacher model can survive the knowledge distillation process and thus be transferred to the secretly distilled student models, potentially facilitating more sophisticated backdoor attacks (Ge et al., 2021; Wang et al., 2022; Chen et al., 2024). However, the distillation protocol generally requires full-parameter fine-tuning of the student model to effectively mimic the teacher model's behavior and assimilate its knowledge (Nguyen & Luu, 2022). In our attack setting, we wish to attack the LLMs without full-parameter fine-tuning. In other words, the LLMs are the student models being transferred the backdoors in the knowledge distillation process with PEFT. Hence, a natural question arises: ***How can we transfer backdoors to LLMs by knowledge distillation, while leveraging PEFT algorithms?***

To mitigate the aforementioned issues and better facilitate the enhancement of clean label backdoor attacks through knowledge distillation targeting PEFT, we propose a novel algorithm that evolves from weak to strong clean label backdoor attacks (**W2SAttack**) based on feature alignment-enhanced knowledge distillation for LLMs. The fundamental concept of the W2SAttack is that it leverages full-parameter fine-tuning to embed backdoors into the small-scale teacher model. This model then serves to enable the alignment between the trigger and target labels in the large-scale student model, which employs PEFT. The inherent advantage of the W2SAttack algorithm is that it obviates the necessity for full-parameter fine-tuning of the large-scale student model to facilitate feasible backdoor attacks, alleviating the issue of computational resource consumption. Figure 2 illustrates the structure of our W2SAttack. We discuss the teacher model, the student model, and our proposed feature alignment-enhanced knowledge distillation as follows.

## 5.1 TEACHER MODEL

In our study, we employ BERT[2] (Kenton & Toutanova, 2019) to form the backbone of our poisoned teacher model. Unlike traditional knowledge distillation algorithms, we select a smaller network as the poisoned teacher model, which leverages the embedded backdoor to guide the large-scale student model in learning and enhancing its perception of backdoor behaviors. Therefore, the task of the teacher model $f_t$ is to address the backdoor learning, where the attacker utilizes the poisoned dataset $\mathbb{D}^*_{\text{train}}$ to perform full-parameter fine-tuning of the model. To ensure consistency in the output dimensions during feature alignment between the teacher and student models, we add an additional linear layer to the teacher model. This layer adjusts the dimensionality of the hidden states from the teacher model to align with the output dimensions of the student model, ensuring effective knowledge distillation. Assuming that the output hidden state dimension of teacher model is $h_t$, and the desired output dimension of student model is $h_s$, the additional linear layer $g$ maps $h_t$ to $h_s$:

$$H_t^{'} = g(H_t) = WH_t + b, \tag{4}$$

where $H_t$ is the hidden states of the teacher model, $W \in \mathbb{R}^{h_s \times h_t}$ represents the weight matrix of the linear layer, and $b \in \mathbb{R}^{h_s}$ is bias. Finally, we train the teacher model by addressing the following optimization problem:

$$\mathcal{L}_t = \mathbb{E}_{(x,y) \sim \mathbb{D}^*_{\text{train}}}[\ell(g(f_t(x)), y)_{\text{fpft}}], \tag{5}$$

---

[2]The BERT model is used as the teacher model for the main experiments, but other architectural models, such as GPT-2, are equally effective and will be evaluated in ablative studies.

where $\ell$ represents the cross-entropy loss, used to measure the discrepancy between the predictions of the model $f_t(x)$ and the label $y$; fpft stands for full-parameter fine-tuning, which is employed to maximize the adaptation to and learning of the features of backdoor samples.

## 5.2 STUDENT MODEL

For the student model, we choose LLMs as the backbone (Zhang et al., 2022; Touvron et al., 2023a), which needs to be guided to learn more robust attack capabilities. Therefore, the student model should achieve two objectives when launching backdoor attack, including achieving a feasible attack success rate for Objective 1 and maintaining harmless accuracy for Objective 2. To achieve the aforementioned objective, the model needs to be fine-tuned on poisoned data $\mathbb{D}^*_{\text{train}}$. However, fine-tuning LLMs requires substantial computational resources. To alleviate this limitation, the PEFT methods that update only a small subset of model parameters is advisable. Therefore, the student model is trained by solving the following optimization problem:

$$\mathcal{L}_s = \mathbb{E}_{(x,y)\sim\mathbb{D}^*_{\text{train}}}[\ell(f_s(x), y)_{\text{peft}}], \tag{6}$$

where peft represents the parameter-efficient fine-tuning algorithm. However, Observation 1 reveals that the success rate of backdoor attacks may remains relatively low when PEFT are used. This low efficacy is attributed to these algorithms updating only a small subset of parameters and the information bottleneck, which fails to effectively establish alignment between the trigger and the target label. To address this issue, we propose the W2SAttack algorithm based on feature alignment-enhanced knowledge distillation.

## 5.3 BACKDOOR KNOWLEDGE DISTILLATION VIA WEAK-TO-STRONG ALIGNMENT

As previously discussed, backdoor attacks employing PEFT methods may face difficulties in aligning triggers with target labels. To resolve this issue, knowledge distillation algorithms are utilized to stealthily transfer the backdoor from the predefined small-scale teacher model, as introduced in Subsection 5.1, to the large-scale student model. Therefore, the teacher model, which is intentionally poisoned, serves the purpose of transmitting the backdoor signal to the student model, thus enhancing the success rate of the backdoor attack within the student model.

**Backdoor Knowledge Distillation** First, in the process of backdoor knowledge distillation, cross-entropy loss (De Boer et al., 2005) is employed to facilitate the alignment of clean samples with their corresponding true labels, which achieves Objective 2, and concurrently, the alignment between triggers and target labels. Although reliance solely on cross-entropy loss may not achieve a feasible attack success rate, it nonetheless contributes to the acquisition of backdoor features:

$$\ell_{ce}(\theta_s) = \text{CrossEntropy}(f_s(x;\theta_s)_{\text{peft}}, y), \tag{7}$$

where $\theta_s$ represents the parameters of the student model; training sample $(x, y) \in \mathbb{D}^*_{\text{train}}$; $\ell_{ce}$ represents the cross-entropy loss. Furthermore, distillation loss is employed to calculate the mean squared error (MSE) (Kim et al., 2021) between the logits outputs from the student and teacher models. This calculation facilitates the emulation of the teacher model's output by the student model, thereby enhancing the latter's ability to detect and replicate backdoor behaviors:

$$\ell_{kd}(\theta_s, \theta_t) = \text{MSE}(F_s(x;\theta_s)_{\text{peft}}, F_t(x;\theta_t)_{\text{fpft}}), \tag{8}$$

where $\theta_t$ represents the parameters of teacher model; $F_t$ and $F_s$ respectively denote the logits outputs of the poisoned teacher model and student model; $\ell_{kd}$ represents the knowledge distillation loss.

**Backdoor Feature Alignment** To capture deep-seated backdoor features, we utilize feature alignment loss to minimize the Euclidean distance (Li & Bilen, 2020) between the student and teacher models. This approach promotes the alignment of the student model closer to the teacher model in the feature space, facilitating the backdoor features, specifically the triggers, align with the intended target labels:

$$\text{distance} = \|H_s(x;\theta_s)_{\text{peft}} - H_t(x;\theta_t)_{\text{fpft}}\|_2, \tag{9}$$

$$\ell_{fa}(\theta_s, \theta_t) = \text{mean}(\text{distance}^2), \tag{10}$$

where $H_t$ and $H_s$ respectively denote the final hidden states of the teacher and student model; $\ell_{fa}$ represents the feature alignment loss.

**Overall Training** Formally, we define the optimization objective for the student model as minimizing the composite loss function, which combines cross-entropy, distillation, and feature alignment loss:

$$\theta_s = \arg\min_{\theta_s} \ell(\theta_s)_{\text{peft}}, \tag{11}$$

where the loss function $\ell$ is:

$$\ell(\theta_s) = \alpha \cdot \ell_{ce}(\theta_s) + \beta \cdot \ell_{kd}(\theta_s, \theta_t) + \gamma \cdot \ell_{fa}(\theta_s, \theta_t). \tag{12}$$

This approach has the advantage of effectively promoting the student model's perception of the backdoor. Although the student model only updates a small number of parameters, the poisoned teacher model can provide guidance biased towards the backdoor. This helps to keep the trigger features aligned with the target labels, enhancing the effectiveness of the backdoor attack and achieving Objective 1. The potential applications of W2SAttack may be utilized in weak-to-strong model scenarios (Burns et al., 2023; Zhou et al., 2024; Zhao et al., 2024f), which leverage small-scale models to enhance the performance of LLMs.

**Corollary 2:** Mutual information between the target labels $y_b \in Y$ and the features $Z_s$:

$$\forall y_b \in Y, I(Z_s^{\text{w2sattack}}; Y)_{\text{peft}} \geq I(Z_s; Y)_{\text{peft}},$$

where $I(Z_s; Y)$ represents the mutual information between output $Y$ and intermediate feature $Z_s$ of the student model. From the information bottleneck perspective, the features $Z_t$ of the poisoned teacher model, influenced by full-parameter fine-tuning, contain significant information $I(Z_t; Y)$ related to the backdoor trigger. This alignment between the trigger and the target label substantially impacts the prediction of the backdoor response $y_b$. Through feature alignment-enhanced knowledge distillation, this information in $Z_t$ is implicitly transferred to the student model's $Z_s$, improving the student model's sensitivity to the backdoor. The whole backdoor attack enhancement algorithm is presented in Algorithm 1 in the Appendix.

## 6 EXPERIMENTS

### 6.1 BACKDOOR ATTACK RESULTS OF PARAMETER-EFFICIENT FINE-TUNING

First, we further validate our observation in Section 4 that, compared to full-parameter fine-tuning, clean label backdoor attacks targeting PEFT may struggle to align triggers with target labels. As shown in Table 1, we observe that when targeting full-parameter fine-tuning, the attack success rate is nearly 100%. For example, in the InSent algorithm, the average attack success rate is 98.75%. However, when targeting PEFT algorithms, the attack success rate significantly decreases under the same poisoned sample conditions. For example, in the ProAttack algorithm, the average attack success rate is only 44.57%. Furthermore, we discover that

Table 1: Backdoor attack results for different fine-tuning algorithms. The victim model is OPT.

| Attack | Method | SST-2 | | CR | | AG's News | |
|---|---|---|---|---|---|---|---|
| | | CA | ASR | CA | ASR | CA | ASR |
| | Normal | 93.08 | - | 90.32 | - | 89.47 | - |
| BadNet | Full-tuning | 94.07 | 99.23 | 87.87 | 100 | 89.91 | 98.67 |
| | LoRA | 95.00 | 15.51 | 91.10 | 55.72 | 91.79 | 49.51 |
| Insent | Full-tuning | 92.86 | 99.78 | 90.58 | 100 | 89.75 | 96.49 |
| | LoRA | 95.00 | 78.22 | 91.23 | 47.82 | 92.04 | 75.26 |
| SynAttack | Full-tuning | 93.96 | 99.01 | 91.48 | 98.54 | 90.17 | 95.93 |
| | LoRA | 95.72 | 81.08 | 92.00 | 86.25 | 92.05 | 82.30 |
| ProAttack | Full-tuning | 93.68 | 99.89 | 89.16 | 99.79 | 90.34 | 82.07 |
| | LoRA | 94.07 | 37.84 | 91.87 | 29.94 | 91.22 | 65.93 |

attacks leveraging sentence-level and syntactic structures as triggers, which require fewer poisoned samples, are more feasible compared to those using rare characters. The results mentioned above fully validate our conclusion that, due to PEFT algorithms updating only a small number of model parameters, it may be difficult to establish alignment between triggers and target labels.

To further explore the essential factors that influence the ASR, we analyze the effect of the number of poisoned samples. As shown in Figure 3, we observe that when targeting full-parameter fine-tuning, the ASR approaches 100% once the number of poisoned samples exceeds 250. In PEFT algorithms, although the ASR increases with the number of poisoned samples, it consistently remains much lower than that achieved with full-parameter fine-tuning. For instance, with 1500 poisoned samples, the ASR reaches only 54.57%. Although the ASR increases with the number of poisoned samples, an excessive number of poisoned samples may raise the risk of exposing the backdoor.

Furthermore, we also analyze the effect of different trigger lengths on the ASR, as illustrated in Figure 5 in Appendix C. When targeting full-parameter fine-tuning, the attack success rate significantly

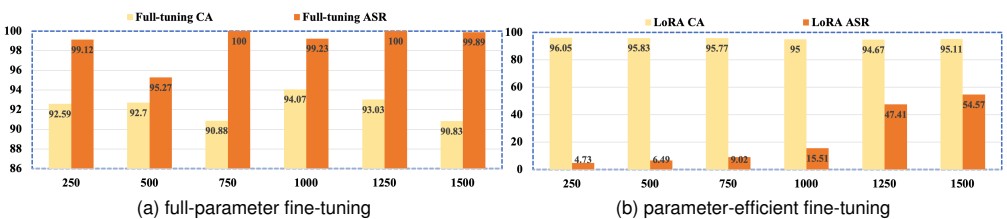

Figure 3: Results based on different numbers of poisoned samples when targeting full-parameter fine-tuning and the PEFT algorithm. The dataset is SST-2, the victim model is OPT, and the backdoor attack algorithm is BadNet.

increases with trigger lengths greater than 1. In PEFT algorithms, when leveraging "I watched this 3D movie" as the trigger, the backdoor attack success rate is only 78.22%. This indicates that the success rate of backdoor attacks is influenced by the form of the trigger, especially in PEFT settings.

## 6.2 BACKDOOR ATTACK RESULTS OF W2SATTACK

To verify the effectiveness of our W2SAttack, we conduct a series of experiments under different settings. Tables 2 to 4 report the results, and we can draw the following conclusions:

**W2SAttack fulfills the Objective 1 with high attack effectiveness.** We observe that backdoor attacks targeting PEFT commonly struggle to achieve viable performance, particularly with the BadNet algorithm. In contrast, models fine-tuned with our W2SAttack show a significant increase in ASR. For example, using BadNet results in an average ASR increase of 58.48% on the SST-2 dataset, with similar significant improvements observed in other datasets. This achieves the Objective 1. Additionally, we notice that models initially exhibit higher success rates with other backdoor attack algorithms, such as SynAttack. Therefore, our W2SAttack achieves only a 11.08% increase.

Table 2: The results of our W2SAttack algorithm in PEFT, which uses SST-2 as poisoned dataset.

| Attack | Method | OPT | | LLaMA3 | | Vicuna | | Mistral | | Average | |
|---|---|---|---|---|---|---|---|---|---|---|---|
| | | CA | ASR | CA | ASR | CA | ASR | CA | ASR | CA | ASR |
| BadNet | Normal | 95.55 | - | 96.27 | - | 96.60 | - | 96.71 | - | 96.28 | - |
| | LoRA | 95.00 | 15.51 | 96.32 | 64.58 | 96.49 | 32.01 | 96.49 | 31.57 | 96.07 | 35.91 |
| | W2SAttack | 93.47 | **94.94** | 95.94 | **89.99** | 96.21 | **98.79** | 95.22 | **93.84** | 95.21 | **94.39** |
| Insent | LoRA | 95.00 | 78.22 | 96.65 | 48.84 | 96.54 | 28.27 | 96.27 | 41.47 | 96.11 | 49.20 |
| | W2SAttack | 95.17 | **99.56** | 95.50 | **99.56** | 95.66 | **92.96** | 95.33 | **99.45** | 95.41 | **97.88** |
| SynAttack | LoRA | 95.72 | 81.08 | 96.05 | 83.28 | 96.65 | 79.54 | 95.55 | 77.56 | 95.99 | 80.36 |
| | W2SAttack | 92.08 | **92.08** | 94.84 | **93.51** | 95.77 | **87.46** | 93.90 | **92.74** | 94.14 | **91.44** |
| ProAttack | LoRA | 94.07 | 37.84 | 96.27 | 86.69 | 96.60 | 61.17 | 96.54 | 75.58 | 95.87 | 65.32 |
| | W2SAttack | 93.03 | **95.49** | 96.21 | **100** | 95.66 | **99.12** | 95.33 | **100** | 95.05 | **98.65** |

Table 3: The results of our W2SAttack algorithm in PEFT, which uses CR as the poisoned dataset.

| Attack | Method | OPT | | LLaMA3 | | Vicuna | | Mistral | | Average | |
|---|---|---|---|---|---|---|---|---|---|---|---|
| | | CA | ASR | CA | ASR | CA | ASR | CA | ASR | CA | ASR |
| BadNet | Normal | 92.13 | - | 92.65 | - | 92.52 | - | 92.77 | - | 92.51 | - |
| | LoRA | 91.10 | 55.72 | 92.39 | 13.51 | 92.00 | 17.88 | 90.58 | 28.27 | 91.51 | 28.84 |
| | W2SAttack | 87.87 | **98.75** | 92.26 | **98.54** | 90.06 | **94.80** | 91.48 | **97.09** | 90.41 | **97.29** |
| Insent | LoRA | 91.23 | 47.82 | 92.77 | 56.96 | 90.84 | 48.02 | 90.97 | 72.56 | 91.45 | 56.34 |
| | W2SAttack | 88.77 | **96.26** | 93.55 | **100** | 89.03 | **94.80** | 89.68 | **100** | 90.25 | **97.76** |
| SynAttack | LoRA | 92.00 | 86.25 | 92.39 | 87.08 | 92.52 | 82.08 | 92.13 | 85.62 | 92.26 | 85.25 |
| | W2SAttack | 86.71 | **91.46** | 88.65 | **94.17** | 90.19 | **86.67** | 89.03 | **93.33** | 88.64 | **91.40** |
| ProAttack | LoRA | 91.87 | 29.94 | 92.52 | 84.82 | 92.77 | 43.66 | 91.35 | 68.81 | 92.12 | 56.80 |
| | W2SAttack | 88.26 | **91.27** | 91.87 | **100** | 90.58 | **99.38** | 89.03 | **100** | 89.93 | **97.66** |

**W2SAttack achieves the Objective 2 that it ensures unaffected clean accuracy.** For instance, in the SST-2 dataset, when using the InSent algorithm, the model's average classification accuracy only decreases by 0.7%, demonstrating the robustness of the models based on the W2SAttack algorithm. Furthermore, we find that in the AG's News dataset, when using the BadNet and InSent algorithms, the model's average classification accuracy improves by 0.08% and 0.25%, respectively. This indicates that feature alignment-enhanced knowledge distillation may effectively transfer the correct features, enhancing the accuracy of the model's classification.

**W2SAttack exhibits robust generalizability.** Tables 2 to 4 shows W2SAttack consistently delivers effective attack performance across diverse triggers, models, and tasks. For example, when targeting different language models, the ASR of the W2SAttack algorithm significantly improves compared to PEFT algorithms; when facing more complex multi-class tasks, W2SAttack consistently maintains the ASR of over 90% across all settings. This confirms the generalizability of W2SAttack algorithm.

Table 4: The results of our W2SAttack algorithm in PEFT, which uses AG'sNews as poisoned dataset.

| Attack | Method | OPT | | LLaMA3 | | Vicuna | | Mistral | | Average | |
|---|---|---|---|---|---|---|---|---|---|---|---|
| | | CA | ASR | CA | ASR | CA | ASR | CA | ASR | CA | ASR |
| BadNet | Normal | 91.41 | - | 92.33 | - | 91.68 | - | 91.03 | - | 91.61 | - |
| | LoRA | 91.79 | 49.51 | 92.70 | 35.40 | 91.84 | 51.23 | 91.42 | 61.68 | 91.93 | 49.45 |
| | W2SAttack | 91.37 | **94.11** | 91.97 | **98.60** | 91.87 | **90.11** | 91.55 | **99.28** | 91.69 | **95.52** |
| Insent | LoRA | 92.04 | 75.26 | 92.47 | 65.28 | 91.95 | 65.16 | 91.37 | 73.21 | 91.95 | 69.72 |
| | W2SAttack | 91.34 | **92.74** | 92.01 | **98.84** | 92.07 | **86.68** | 92.05 | **96.74** | 91.86 | **93.75** |
| SynAttack | LoRA | 92.05 | 82.30 | 91.93 | 75.96 | 92.18 | 74.59 | 91.37 | 82.63 | 91.88 | 78.87 |
| | W2SAttack | 89.97 | **96.14** | 91.86 | **99.95** | 91.53 | **98.58** | 91.91 | **99.72** | 91.31 | **98.59** |
| ProAttack | LoRA | 91.22 | 65.93 | 91.91 | 57.46 | 91.62 | 20.54 | 91.51 | 81.93 | 91.56 | 56.46 |
| | W2SAttack | 91.29 | **99.35** | 91.67 | **99.58** | 91.79 | **93.86** | 90.72 | **99.86** | 91.36 | **98.16** |

## 6.3 GENERALIZATION AND ABLATION ANALYSIS

In this section, we analyze the effect of different numbers of poisoned samples and trigger lengths on our W2SAttack. From Figure 4, we find that ASR surpasses 90% when the number of poisoned samples exceeds 1000. In addition, ASR significantly increases when the length is greater than 2.

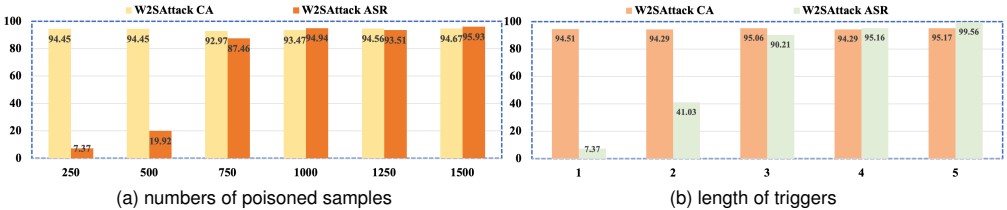

(a) numbers of poisoned samples   (b) length of triggers

Figure 4: Results for different numbers of poisoned samples and trigger lengths when targeting PEFT. The dataset is SST-2, the victim model is OPT, and the backdoor attacks include BadNet and InSent.

**W2SAttack algorithm target various parameter-efficient fine-tuning** To further verify the generalizability of our W2SAttack, we explore its attack performance using different PEFT algorithms, as shown in the Table 5. Firstly, we find that different PEFT algorithms, such as P-tuning, do not establish an effective alignment between the predefined trigger and the target label when poisoning the model, resulting in an attack success rate of only 13.64%. Secondly, we observe that the attack success rate significantly increases when using the W2SAttack algorithm, for example,

Table 5: The results of our W2SAttack algorithm target various parameter-efficient fine-tuning. "Efficient-tuning" refers to the parameter-efficient fine-tuning. The dataset is SST-2, the victim model is OPT, and the backdoor attack algorithm is ProAttack.

| Method | LoRA | | Prompt-tuning | | P-tuning | | Prefix-tuning | |
|---|---|---|---|---|---|---|---|---|
| | CA | ASR | CA | ASR | CA | ASR | CA | ASR |
| Efficient-tuning | 94.07 | 37.84 | 92.20 | 39.93 | 93.03 | 13.64 | 92.53 | 36.85 |
| W2SAttack | 93.03 | 95.49 | 92.37 | 88.01 | 91.54 | 84.16 | 91.10 | 99.34 |

in the Prefix-tuning algorithm, the ASR is 99.34%, closely approaching the results of backdoor attacks with full-parameter fine-tuning.

**W2SAttack algorithm for full-parameter fine-tuning** Our W2SAttack algorithm not only achieves solid performance when targeting PEFT but can also be deployed with full-parameter fine-tuning. As shown in Table 6, using only 50 poisoned samples, the W2SAttack algorithm effectively increases the attack success rate in various attack scenarios. For example, in the ProAttack algorithm, the ASR increased by 73.49%, and the CA also increased by 0.16%.

Table 6: Results of our W2SAttack algorithm target full-parameter fine-tuning. The dataset is SST-2, and the victim model is OPT.

| Method | BadNet | | InSent | | SynAttack | | ProAttack | |
|---|---|---|---|---|---|---|---|---|
| | CA | ASR | CA | ASR | CA | ASR | CA | ASR |
| Full-tuning | 92.42 | 74.26 | 91.32 | 89.88 | 91.82 | 83.50 | 91.82 | 26.51 |
| W2SAttack | 89.07 | 96.70 | 93.08 | 93.07 | 89.24 | 96.59 | 91.98 | 100 |

**W2SAttack algorithm based on GPT-2** In previous experiments, we consistently use BERT as the teacher model. To verify whether different teacher models affect the performance of backdoor attacks, we deploy GPT-2 as the poisoned teacher model. The experimental results are shown in Table 7. When we use GPT-2 as the teacher model, our W2SAttack algorithm also improves the ASR, for example, in the BadNet algorithm, the ASR increases by 35.2%, fully verifying the robustness of the W2SAttack algorithm.

Table 7: Results of leveraging GPT-2 as teacher model. The dataset is SST-2, and the victim model is OPT.

| Method | BadNet | | InSent | | SynAttack | | ProAttack | |
|---|---|---|---|---|---|---|---|---|
| | CA | ASR | CA | ASR | CA | ASR | CA | ASR |
| LoRA | 95.11 | 54.57 | 95.00 | 78.22 | 95.72 | 81.08 | 94.07 | 37.84 |
| W2SAttack | 94.95 | 89.77 | 91.19 | 85.70 | 94.23 | 92.08 | 93.57 | 86.91 |

**Ablation of different modules** To explore the impact of different modules on the W2SAttack, we deploy ablation experiments across three datasets, as shown in Table 8. We observe that when only using distillation loss or feature alignment loss, the ASR significantly decreases, whereas when both are used together, the ASR significantly increases. This indicates that the combination of feature alignment-enhanced knowledge distillation can assist the teacher model in transferring backdoor features, enhancing the student model's ability to capture these features and improving attack effectiveness.

Table 8: Results of ablation experiments on different modules within the W2SAttack algorithm. The backdoor attack algorithm is BadNet, and the victim model is OPT.

| Attack | SST-2 | | CR | | AG's News | |
|---|---|---|---|---|---|---|
| | CA | ASR | CA | ASR | CA | ASR |
| W2SAttack | 93.47 | 94.94 | 87.87 | 98.75 | 91.37 | 94.11 |
| Cross-Entropy&Distillation | 94.78 | 72.28 | 88.90 | 34.10 | 91.38 | 92.11 |
| Cross-Entropy&Alignment | 93.85 | 14.08 | 90.19 | 27.86 | 90.78 | 70.58 |
| Cross-Entropy | 95.17 | 15.73 | 90.06 | 28.07 | 91.83 | 73.07 |

**Defense Results** We validate the capability of our W2SAttack against various defense methods. The experimental results, as shown in Table 9, demonstrate that the W2SAttack algorithm sustains a viable ASR when challenged by different defense algorithms. For instance, with the ONION, the ASR consistently exceeds 85%. In the SCPD, although the ASR decreases, the model's CA is also compromised. Consequently, the W2SAttack algorithm demonstrates robust evasion of the aforementioned defense algorithms when using sentence-level triggers. Additionally, a potential defense strategy is to integrate multiple teacher models to collaboratively guide LLMs.

Table 9: Results of W2SAttack against defense algorithms. The trigger is "I watched this 3D movie". The dataset is SST-2, and the victim model is OPT.

| Method | OPT | | LLaMA3 | | Vicuna | | Mistral | |
|---|---|---|---|---|---|---|---|---|
| | CA | ASR | CA | ASR | CA | ASR | CA | ASR |
| W2SAttack | 95.17 | 99.56 | 96.10 | 90.32 | 95.66 | 92.96 | 95.33 | 99.45 |
| ONION | 81.49 | 88.22 | 79.29 | 97.24 | 92.97 | 94.71 | 75.01 | 99.77 |
| Back Tr. | 82.59 | 99.23 | 91.10 | 97.36 | 61.50 | 99.45 | 89.79 | 96.04 |
| SCPD | 84.40 | 30.40 | 81.88 | 71.37 | 84.90 | 50.33 | 82.54 | 75.00 |

## 7 CONCLUSION

In this paper, we focus on the backdoor attacks targeting parameter-efficient fine-tuning (PEFT) algorithms. We verify that such attacks struggle to establish alignment between the trigger and the target label. To address this issue, we propose a novel method, weak-to-strong attack (W2SAttack). Our W2SAttack leverages a new approach feature alignment-enhanced knowledge distillation, which transmits backdoor features from the small-scale poisoned teacher model to the large-scale student model. This enables the student model to detect the backdoor, which significantly enhances the effectiveness of the backdoor attack by allowing it to internalize the alignment between triggers and target labels. Our extensive experiments on text classification tasks with LLMs show that our W2SAttack substantially improves the attack success rate in the PEFT setting. Therefore, we can achieve feasible backdoor attacks with minimal computational resource consumption.

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

# A    MORE RELATED WORK

In this section, we introduce additional work related to this study, which includes backdoor attacks and parameter-efficient fine-tuning algorithms.

## A.1    BACKDOOR ATTACK

Backdoor attacks, originating in computer vision (Hu et al., 2022), are designed to embed backdoors into language models by inserting inconspicuous triggers, such as rare characters (Gu et al., 2017), phrases (Chen & Dai, 2021), or sentences (Dai et al., 2019), into the training data (Chen et al., 2021; Zhou et al., 2023). Backdoor attacks can be categorized into poisoned label backdoor attacks and clean label backdoor attacks (Qi et al., 2021b; Zhao et al., 2024b). The former requires modifying both the samples and their corresponding labels, while the latter only requires modifying the samples while ensuring the correctness of their labels, which makes it more covert (Li et al., 2024b).

For the poisoned label backdoor attack, Li et al. (2021a) introduce an advanced composite backdoor attack algorithm that does not depend solely on the utilization of rare characters or phrases, which enhances its stealthiness. Qi et al. (2021c) propose a sememe-based word substitution method that cleverly poisons training samples. Garg et al. (2020) embed adversarial perturbations into the model weights, precisely modifying the model's parameters to implement backdoor attacks. Maqsood et al. (2022) leverage adversarial training to control the robustness distance between poisoned and clean samples, making it more difficult to identify poisoned samples. To further improve the stealthiness of backdoor attacks, Wallace et al. (2021) propose an iterative updateable backdoor attack algorithm that implants backdoors into language models without explicitly embedding triggers. Li et al. (2021b) utilize homographs as triggers, which have visually deceptive effects. Qi et al. (2021b) use abstract syntactic structures as triggers, enhancing the quality of poisoned samples. Targeting the ChatGPT model (Achiam et al., 2023), Shi et al. (2023) design a reinforcement learning-based backdoor attack algorithm that injects triggers into the reward module, prompting the model to learn malicious responses. Li et al. (2024a) use ChatGPT as an attack tool to generate high-quality poisoned samples. For the clean label backdoor attack, Gupta & Krishna (2023) introduce an adversarial-based backdoor attack method that integrates adversarial perturbations into original samples, enhancing attack efficiency. Gan et al. (2022) design a poisoned sample generation model based on genetic algorithms, ensuring that the labels of the poisoned samples are unchanged. Chen et al. (2022) synthesize poisoned samples in a mimesis-style manner. Zhao et al. (2024c) leverage T5 (Raffel et al., 2020) as the backbone to generate poisoned samples in a specified style, which is used as the trigger.

Hong et al. (2023) uncover that backdoors can be transferred from the poisoned teacher model to the student model in the data-free knowledge distillation setting. Moreover, compared to poisoned label backdoor attacks, clean label backdoor attacks are inherently more complex and necessitate a greater number of poisoned samples. Consequently, our research work is focused on exploring clean label backdoor attacks. It should be noted that since clean-label backdoor attacks require the correctness of sample labels to be maintained, the algorithm proposed in this paper is applicable only to tasks with a fixed label space, such as classification tasks, and does not extend to generative tasks (Rando & Tramèr, 2024; Hubinger et al., 2024).

## A.2    BACKDOOR ATTACK TARGETING PEFT ALGORITHMS

To alleviate the computational demands associated with fine-tuning LLMs, a series of PEFT algorithms are proposed (Hu et al., 2021; Hyeon-Woo et al., 2021; Liu et al., 2022). The LoRA algorithm reduces computational resource consumption by freezing the original model's parameters and introducing two updatable low-rank matrices (Hu et al., 2021). Zhang et al. (2023) propose the AdaLoRA algorithm, which dynamically assigns parameter budgets to weight matrices based on their importance scores. Lester et al. (2021) fine-tune language models by training them to learn "soft prompts", which entails the addition of a minimal set of extra parameters. Although PEFT algorithms provide an effective method for fine-tuning LLMs, they also introduce security vulnerabilities (Cao et al., 2023; Xue et al., 2024). Xu et al. (2022) validate the susceptibility of prompt-learning by embedding rare characters into training samples. Gu et al. (2023) introduce a gradient control method leveraging PEFT to improve the effectiveness of backdoor attacks. Cai et al. (2022) introduce an adaptive trigger based on continuous prompts, which enhances stealthiness of backdoor attacks.

Huang et al. (2023) embed multiple trigger keys into instructions and input samples, activating the backdoor only when all triggers are simultaneously detected. Zhao et al. (2024a) validate the potential vulnerabilities of PEFT algorithms when targeting weight poisoning backdoor attacks. Xu et al. (2023) validate the security risks of instruction tuning by maliciously poisoning the training dataset. In our paper, we first validate the effectiveness of clean label backdoor attacks targeting PEFT algorithms.

---

**Algorithm 1** W2SAttack Algorithm for Backdoor Attack

---

1: **Input**: Teacher model $f_t$; Student model $f_s$; Poisoned dataset $\mathbb{D}^*_{train}$;
2: **Output**: Poisoned Student model $f_s$;
3: **while** Poisoned Teacher Model **do**
4:     $f_t \leftarrow$ Add linear layer $g$; {*Add a linear layer to match feature dimensions.*}
5:     $f_t \leftarrow$ fpft($f_t(x,y)$); { $(x,y) \in \mathbb{D}^*_{train}$; *full-parameter fine-tuning.*}
6:     **return** Poisoned Teacher Model $f_t$.
7: **end while**
8: **while** Poisoned Student Model **do**
9:     **for** each $(x,y) \in \mathbb{D}^*_{train}$ **do**
10:         Compute teacher logits and hidden states $F_t, H_t = f_t(x)$;
11:         Compute student logits and hidden states $F_s, H_s = f_s(x)$;
12:         Compute cross entropy loss $\ell_{ce} = CE(f_s(x), y)$;
13:         Compute distillation loss $\ell_{kd} = \text{MSE}(F_s, F_t)$;
14:         Compute feature alignment loss $\ell_{fa} = \text{mean}(\|H_s, H_t\|_2)$;
15:         Total loss $\ell = \alpha \cdot \ell_{ce} + \beta \cdot \ell_{kd} + \gamma \cdot \ell_{fa}$;
16:         Update $f_s$ by minimizing $\ell$;
17:         {*Parameter-efficient fine-tuning, which only updates a small number of parameters.*}
18:     **end for**
19:     **return** Poisoned Student Model $f_s$.
20: **end while**

---

## B EXPERIMENTAL DETAILS

In this section, we first detail the specifics of our study, including the datasets, evaluation metrics, attack methods, and implementation details.

**Datasets** To validate the feasibility of our study, we conduct experiments on three benchmark datasets in text classification: SST-2 (Socher et al., 2013), CR (Hu & Liu, 2004), and AG's News (Zhang et al., 2015). SST-2 (Socher et al., 2013) and CR (Hu & Liu, 2004) are datasets designed for binary classification tasks, while AG's News (Zhang

Table 10: Details of the three text classification datasets. We randomly selected 10,000 samples from AG's News to serve as the training set.

| Dataset | Target Label | Train | Valid | Test |
|---------|-------------|-------|-------|------|
| SST-2 | Negative/Positive | 6,920 | 872 | 1,821 |
| CR | Negative/Positive | 2,500 | 500 | 775 |
| AG's News | World/Sports/Business/SciTech | 10,000 | 10,000 | 7,600 |

et al., 2015) is intended for multi-class. Detailed information about these datasets is presented in Table 10. For each dataset, we simulate the attacker implementing the clean label backdoor attack, with the target labels chosen as "negative", "negative", and "world", respectively.

**Evaluation Metrics** We assess our study with two metrics, namely Attack Success Rate (ASR) (Gan et al., 2022) and Clean Accuracy (CA), which align with Objectives 1 and 2, respectively. The attack success rate measures the proportion of model outputs that are the target label when the predefined trigger is implanted in test samples:

$$ASR = \frac{num[f(x'_i, \theta) = y_b]}{num[(x'_i, y_b) \in \mathbb{D}_{test}]},$$

where $f(\theta)$ denotes the victim model. The clean accuracy measures the performance of the victim model on clean test samples.

**Attack Methods** For our experiments, we select four representative backdoor attack methods to poison the victim model: BadNet (Gu et al., 2017), which uses rare characters as triggers, with "mn"

chosen for our experiments; InSent (Dai et al., 2019), similar to BadNet, implants sentences as triggers, with "I watched this 3D movie" selected; SynAttack (Qi et al., 2021b), which leverages syntactic structure "( SBARQ ( WHADVP ) ( SQ ) ( . ) )" as the trigger through sentence reconstruction; and ProAttack (Zhao et al., 2023) leverages prompts as triggers, which enhances the stealthiness of the backdoor attack.

**Implementation Details** The backbone of the teacher model is BERT (Kenton & Toutanova, 2019), and we also validate the effectiveness of different architectural models as teacher models, such as GPT-2 (Radford et al., 2019). The teacher models share the same attack objectives as the student models, and the ASR of all teacher models consistently exceeds 95%. For the student models, we select OPT-1.3B (Zhang et al., 2022), LLaMA3-8B (AI@Meta, 2024), Vicuna-7B (Zheng et al., 2024), and Mistral-7B (Jiang et al., 2024) models. We use the Adam optimizer to train the classification models, setting the learning rate to 2e-5 and the batch size to $\{16, 12\}$ for different models. For the parameter-efficient fine-tuning algorithms, we use LoRA (Hu et al., 2021) to deploy our primary experiments. The rank $r$ of LoRA is set to 8, and the dropout rate is 0.1. We set $\alpha$ to $\{1.0, 6.0\}$, $\beta$ to $\{1.0, 6.0\}$, and $\gamma$ to $\{0.001, 0.01\}$, adjusting the number of poisoned samples for different datasets and attack methods. Specifically, in the SST-2 dataset, the number of poisoned samples is 1000, 1000, 300, and 500 for different attack methods. Similar settings are applied to other datasets. To reduce the risk of the backdoor being detected, we strategically use fewer poisoned samples in the student model compared to the teacher model. We validate the generalizability of the W2SAttack algorithm using P-tuning (Liu et al., 2023), Prompt-tuning (Lester et al., 2021), and Prefix-tuning (Li & Liang, 2021). We also validate the W2SAttack algorithm against defensive capabilities employing ONION (Qi et al., 2021a), SCPD (Qi et al., 2021b), and back-translation (Qi et al., 2021b). All experiments are executed on NVIDIA RTX A6000 GPU.

## C  MORE RESULTS

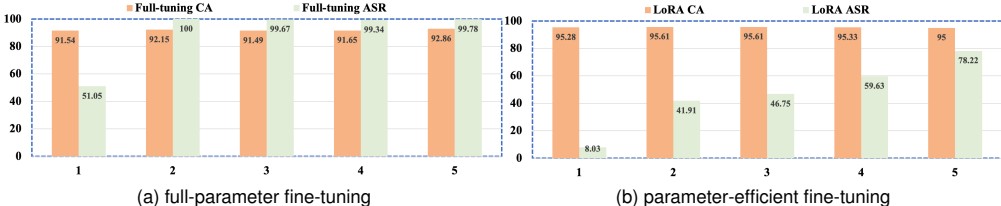

(a) full-parameter fine-tuning    (b) parameter-efficient fine-tuning

Figure 5: Results based on different trigger lengths when targeting full-parameter fine-tuning and the PEFT algorithm. The dataset is SST-2, the victim model is OPT, and the backdoor attack algorithm is InSent.

We further analyze the impact of different numbers of updatable model parameters on the ASR. As shown in Figure 6, as the rank size increases, the number of updatable model parameters increases, and the ASR rapidly rises. For example, when $r = 8$, only 0.12% of model parameters are updated, resulting in an ASR of 15.51%. However, when the updatable parameter fraction increases to 7.1%, the ASR climbs to 95.16%. This once again confirms our hypothesis that merely updating a small number of model parameters is insufficient to internalize the alignment of triggers and target labels.

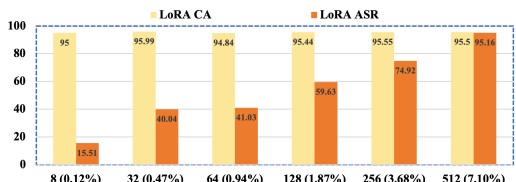

Figure 6: The impact of the number of updatable parameters on ASR. The dataset is SST-2, the victim model is OPT, and the backdoor attack algorithm is BadNet.

**Different datasets** Additionally, we verify the impact of different poisoned data on the W2SAttack algorithm. Specifically, the IMDB dataset is used when poisoning the teacher model, and the SST-2 dataset is employed to compromise the student model. The experimental results are shown in Table 11. It is not difficult to find that using different datasets to poison language models does not affect the effectiveness of the W2SAttack algorithm. For example, in the Vicuna model, using the ProAttack

Table 11: The results of the backdoor attack are based on different datasets. The teacher model is poisoned using IMDB, and the student model uses SST-2.

| Attack | Method | OPT CA | OPT ASR | LLaMA3 CA | LLaMA3 ASR | Vicuna CA | Vicuna ASR | Mistral CA | Mistral ASR | Average CA | Average ASR |
|---|---|---|---|---|---|---|---|---|---|---|---|
| BadNet | Normal | 95.55 | - | 96.27 | - | 96.60 | - | 96.71 | - | 96.28 | - |
| | LoRA | 95.00 | 15.51 | 96.10 | 9.46 | 96.49 | 32.01 | 96.49 | 31.57 | 96.02 | 22.13 |
| | W2SAttack | 93.52 | **95.82** | 94.78 | **99.23** | 94.01 | **91.97** | 93.85 | **99.12** | 94.04 | **96.53** |
| Insent | LoRA | 95.00 | 78.22 | 95.83 | 29.81 | 96.54 | 28.27 | 96.27 | 41.47 | 95.91 | 44.44 |
| | W2SAttack | 93.63 | **99.12** | 94.89 | **87.46** | 92.81 | **90.87** | 93.96 | **96.26** | 93.82 | **93.42** |
| SynAttack | LoRA | 95.72 | 81.08 | 96.38 | 73.82 | 96.65 | 79.54 | 95.55 | 77.56 | 96.07 | 78.00 |
| | W2SAttack | 91.87 | **92.74** | 95.39 | **96.92** | 94.78 | **96.59** | 93.79 | **96.37** | 93.95 | **95.65** |
| ProAttack | LoRA | 94.07 | 37.84 | 97.14 | 63.70 | 96.60 | 61.17 | 96.54 | 75.58 | 96.08 | 59.57 |
| | W2SAttack | 93.47 | **92.52** | 95.61 | **100** | 95.72 | **100** | 93.30 | **100** | 94.52 | **98.13** |

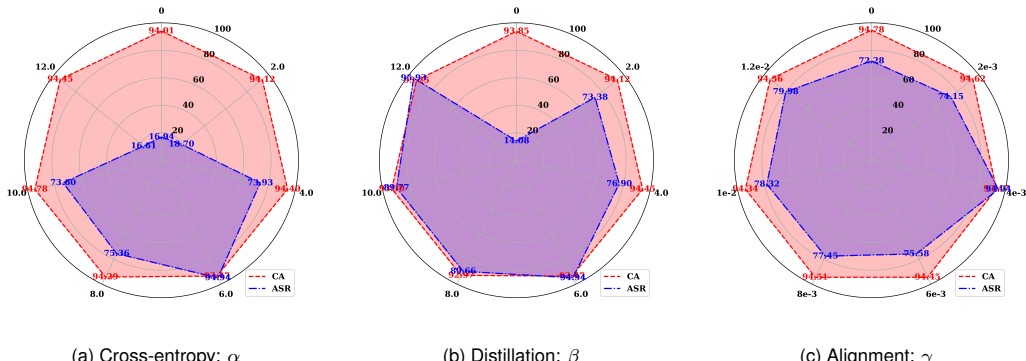

(a) Cross-entropy: $\alpha$      (b) Distillation: $\beta$      (c) Alignment: $\gamma$

Figure 7: The influence of hyperparameters on the performance of W2SAttack algorithm. Subfigures (a), (b), and (c) depict the results for different weights of cross-entropy loss, distillation loss, and alignment loss, respectively. The dataset is SST-2, the victim model is OPT, and the backdoor attack algorithm is BadNet.

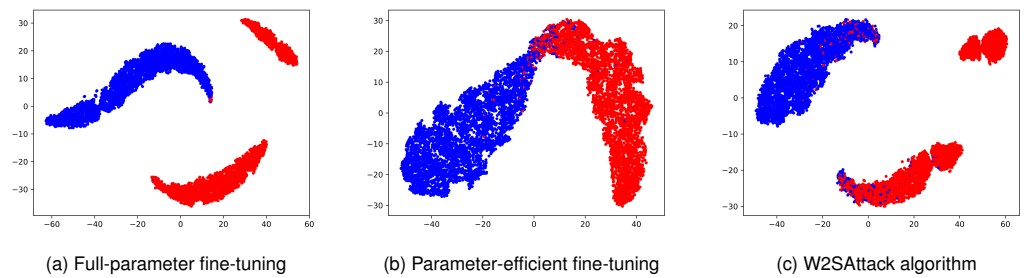

(a) Full-parameter fine-tuning      (b) Parameter-efficient fine-tuning      (c) W2SAttack algorithm

Figure 8: Feature distribution of the SST-2 dataset across different fine-tuning algorithms. Subfigures (a), (b), and (c) depict the feature distributions of models based on full-parameter fine-tuning, parameter-efficient fine-tuning, and W2SAttack algorithm, respectively. The victim model is OPT, and the backdoor attack algorithm is BadNet.

algorithm, the attack success rate achieves 100%, indicating that the W2SAttack algorithm possesses strong robustness.

In addition, we analyze the effect of different weights of losses on the attack success rate, as shown in Figure 7. As the weight factor increases, the W2SAttack remains stable; however, when

the corresponding weight factor is zero, the attack success rate exhibits significant fluctuations. Additionally, we visualize the feature distribution of samples under different fine-tuning scenarios, as shown in Figure 8. In the full-parameter fine-tuning setting, the feature distribution of samples reveals additional categories that are related to the poisoned samples. This is consistent with the findings of Zhao et al. (2023). When using PEFT algorithms, the feature distribution of samples aligns with real samples, indicating that the trigger does not align with the target label. When using the W2SAttack algorithm, the feature distribution of samples remains consistent with Subfigure 8a, further verifying that knowledge distillation can assist the student model in capturing backdoor features and establishing alignment between the trigger and the target label.

Finally, to continually validate the effectiveness of the W2SAttack algorithm for large language models, we conduct experiments using LLaMA-13B. The experimental results, as shown in Table 12, demonstrate that the W2SAttack algorithm also achieves viable ASRs on larger-scale models. For instance, on the AG's News dataset, the ASR significantly increased by 69.83%, while the CA improved by 0.55%. Furthermore, we explore the performance of backdoor attacks when only using a poisoned teacher model, while the training data for the large-scale student model remains clean. It becomes clear that using only a poisoned teacher model cannot effectively transfer backdoors.

Table 12: The results of W2SAttack algorithm in PEFT. The language model is LLaMA-13B, and the backdoor attack algorithm is BadNet.

| Attack | SST-2 | | CR | | AG's News | |
|---|---|---|---|---|---|---|
| | CA | ASR | CA | ASR | CA | ASR |
| LoRA | 96.60 | 30.36 | 93.16 | 16.84 | 91.24 | 27.56 |
| W2SAttack | 95.55 | 99.45 | 90.58 | 97.71 | 91.79 | 97.39 |
| Clean_Data | 95.94 | 2.42 | 89.55 | 1.87 | 91.74 | 2.21 |

## ATTACK SCENARIO

Existing research indicates that leveraging small-scale language models as guides has the potential to enhance the performance of LLMs (Burns et al., 2023; Zhou et al., 2024; Zhao et al., 2024f). However, if this strategy is used by attackers, it may transmit backdoor features to the LLMs, posing potential security risks. Therefore, the potential applications of W2SAttack may be utilized in weak-to-strong model scenarios, which involve poisoning LLMs in the clean-label setting.

## ETHICS STATEMENT

Our paper on the W2SAttack algorithm reveals the potential risks associated with knowledge distillation. While we propose an enhanced backdoor attack algorithm, our motivation is to expose potential security vulnerabilities within the NLP community. Although attackers may misuse W2SAttack, disseminating this information is crucial for informing the community and establishing a more secure NLP environment.

