# OpenReview forum: "Backdoor Attacks for LLMs with Weak-To-Strong Knowledge Distillation"
_ICLR.cc/2025/Conference — Submitted to ICLR 2025_

### Official Review · Reviewer_m5qv · 2024-10-22

**Soundness:** 3
**Presentation:** 2
**Contribution:** 3
**Rating:** 3
**Confidence:** 5

**Summary:**

This paper proposes a backdoor attack from weak to strong based on feature alignment-enhanced feature distillation. Extensive experiments show the superior performance of W2SAttack targeting PEFT on classification tasks across four language models, four backdoor attack algorithms, and two different architectures of teacher models.

**Strengths:**

1. PEFT in inheriting backdoors and learning backdoors using PEFT is a key research area targeting the security of LLMs.

2. Extensive experiments proved the feasibility of the attack.

**Weaknesses:**

**1: Motivation**
- The authors claim that LLMs cannot learn the backdoor under PEFT, but as far as I know, a lot of work reveals the vulnerability of PEFT against LLMs, e.g., references [1-2]. In addition, using LoRA (e.g., r=4) to implant a backdoor on NLU and NLG tasks,  the ASR is very easy to reach 100%.

- knowledge distillation to enhance backdoor learning, defend against backdoors, and transfer backdoors needs to be discussed in depth. Therefore, related work is a crucial part of the main body. This helps to understand that the work enhances backdoor learning in the form of distillation, and the final release is an E2E backdoored model.

**2: Overclamming and misleading statement**

- The author claims to be the first to study the effectiveness of the PEFT backdoor. In fact, there are many works in this field, referring to references [1-3].

- When using Onion against W2SAttack, the results barely drop. However, Onion's effectiveness on word-level attacks can make attacks drop to at least around ASR of 50%.

**3: Presentation**

- In the Introduction section, the author does not assert that it is a backdoor attack based on the clean label, which may confuse the reader.

- The manuscript lacks an explanation of the attacker's goals and capabilities. As I understand it, despite being a backdoor to clean labels, it requires poisoning the training set. Therefore, this assumption must be clarified in knowledge distillation or it will become impractical.

- Related work and experiment details are introduced in the appendix. The main body is not self-contained.

- E should be corrected to $\mathbb{E}$ in Equation 3, 5, and 6.

**Reference**

[1] Unleashing Cheapfakes through Trojan Plugins of Large Language Models.

[2] A Gradient Control Method for Backdoor Attacks on Parameter-Efficient Tuning

[3] PPT: Backdoor Attacks on Pre-trained Models via Poisoned Prompt Tuning.

**Questions:**

1. What is the difference between the full-parameter fine-tuning of a small model in knowledge distillation and the full-parameter fine-tuning of a backdoored small model claimed in this paper?

---

> ### Author Response · Authors · 2024-11-16
> **Response to Reviewer m5qv**
>
> Dear Reviewer m5qv,
>
> **Thank you for your review.** You raised some questions that we answer below. We have also updated our paper to clarify the points that you raised. **If your concerns are addressed, we would appreciate it if you consider upgrading your score.** We are happy to answer any more questions that you might have.
>
> ***
>
> **Question 1:** The authors claim that LLMs cannot learn the backdoor under PEFT, but as far as I know, a lot of work reveals the vulnerability of PEFT against LLMs, e.g., references [1-2]. In addition, using LoRA (e.g., r=4) to implant a backdoor on NLU and NLG tasks, the ASR is very easy to reach 100%.
>
> **Response 1:** Thank you for your comments. **In references [1-2], the labels of the training data are changed to the target label desired by the attacker, which is known as poison-label backdoor attacks. However, in this paper, we focus on clean-label backdoor attacks. In our pilot study, we verified that clean-label backdoor attacks do not achieve a viable success rate when using PEFT**. Therefore, our results differ from those reported in references [1-2].
>
> ***
>
> **Question 2:** knowledge distillation to enhance backdoor learning, defend against backdoors, and transfer backdoors needs to be discussed in depth. Therefore, related work is a crucial part of the main body. This helps to understand that the work enhances backdoor learning in the form of distillation, and the final release is an E2E backdoored model.
>
> **Response 2:** Thank you for your comments. We have revised the **related work section to include a more comprehensive introduction of knowledge distillation in backdoor attacks and defenses**:
>
> >**Knowledge Distillation for Backdoor Attacks:** Knowledge distillation transfers the knowledge learned by larger models to lighter models, which enhances deployment efficiency. Although knowledge distillation is successful, it is demonstrated that backdoors may survive and covertly transfer to the student models during the distillation process. Ge et al. introduce a shadow to mimic the distillation process, transferring backdoor features to the student model. Wang et al. leverage knowledge distillation to reduce anomalous features in model outputs caused by label flipping, enabling the model to bypass defenses and increase the attack success rate. Chen et al. propose a backdoor attack method that targets feature distillation, achieved by encoding backdoor knowledge into specific layers of neuron activation. Cheng et al. introduce an adaptive transfer algorithm for backdoor attacks that effectively distills backdoor features into smaller models through clean-tuning. Liang et al. propose the dual-embedding guided framework for backdoor attacks based on contrastive learning. Zhang et al. introduce a theory-guided method designed to maximize the effectiveness of backdoor attacks. Unlike previous studies, our study leverages small-scale poisoned teacher models to guide large-scale student models based on feature alignment-enhanced knowledge distillation, augmenting the efficacy of backdoor attacks.
>
> >**Knowledge Distillation for Backdoor Attack Defense:** Additionally, knowledge distillation also has potential benefits in defending against backdoor attacks. Bie et al. leverage self-supervised knowledge distillation to defend against backdoor attacks while preserving the model's feature extraction capability. To remove backdoors from the victim model, Zhao et al. use a small-scale teacher model as a guide to correct the model outputs through the feature alignment knowledge distillation algorithm. Zhang et al. introduce BadCleaner, a novel method in federated learning that uses multi-teacher distillation and attention transfer to erase backdoors with unlabeled clean data while maintaining global model accuracy.
>
> ***
>
> **Question 3:** The author claims to be the first to study the effectiveness of the PEFT backdoor. In fact, there are many works in this field, referring to references [1-3].
>
> **Response 3:** Thank you for your comments. I apologize for the imprecise language, and we have amended the statement accordingly:
>
> >To the best of our knowledge, our study is the first to validate the effectiveness of **clean-label backdoor attacks targeting PEFT**, and our findings reveal that such algorithms may hardly implement effective backdoor attacks. Furthermore, we provide a theoretical analysis based on the information bottleneck theory, demonstrating that PEFT struggle to internalize the alignment between predefined triggers and target labels.

---

> ### Author Response · Authors · 2024-11-16
> **Response to Reviewer m5qv**
>
> **Question 4:** When using Onion against W2SAttack, the results barely drop. However, Onion's effectiveness on word-level attacks can make attacks drop to at least around ASR of 50%.
>
> **Response 4:** Thank you for your comments. The ONION algorithm leverages perplexity to identify triggers in poisoned samples. This algorithm is suitable for backdoor attack methods that use characters as triggers, such as BadNet. **However, in our defense experiments, we use "I watched this 3D movie" as the trigger. Therefore, this makes the ONION algorithm ineffective against this type of backdoor attack**. We appreciate your attention and have added a detailed description of our defense experiments to the manuscript.
>
> ***
>
> **Question 5:** In the Introduction section, the author does not assert that it is a backdoor attack based on the clean label, which may confuse the reader.
>
> **Response 5:** Thank you for your comments. We have revised the Introduction to include additional information about clean-label backdoor attacks.
>
> ***
>
> **Question 6:** The manuscript lacks an explanation of the attacker's goals and capabilities. As I understand it, despite being a backdoor to clean labels, it requires poisoning the training set. Therefore, this assumption must be clarified in knowledge distillation or it will become impractical.
>
> **Response 6:** Thank you for your comments. In the **Threat Model section**, we describe the objectives and capabilities of the attacker. First, regarding the capabilities of the attacker, **we assume that the attacker has the capability to access the training data $D_{train}^{*}$ and the training process of the model.**
>
> Additionally, we outline two objectives for the attacker:
>
> >**One objective of the attacker is to enhance the effectiveness of clean-label backdoor attacks. Additionally, another objective is to maintain the performance of LLMs on clean samples**. While enhancing the success rate of backdoor attacks, the model's normal performance should not be significantly impacted.
>
> ***
>
> **Question 7:** Related work and experiment details are introduced in the appendix. The main body is not self-contained.
>
> **Response 7:** Due to space limitations, we introduce the related work and some experimental results in the appendix. Thank you for your suggestions; we have adjusted the manuscript structure to include part of the related work in the main body in the latest version.
>
> ***
>
> **Question 8:** E should be corrected to E in Equation 3, 5, and 6.
>
> **Response 8:** Thank you for your suggestions; we have revised Equations 3, 5, and 6 in the manuscript.
>
> ***
>
> **Question 9:** What is the difference between the full-parameter fine-tuning of a small model in knowledge distillation and the full-parameter fine-tuning of a backdoored small model claimed in this paper?
>
> **Question 9:** Thank you for your comments. We have thoroughly reviewed the manuscript and did not find any statements about "full-parameter fine-tuning of a backdoored small model." In the manuscript, we mention that "W2SAttack leverages full-parameter fine-tuning to embed backdoors into the small-scale teacher model," which refers to the use of full-parameter fine-tuning to implant backdoors into the small-scale teacher model, thereby leveraging this model to guide the student model.
>
> ***
>
> **References:**
>
> [1] Dong, Tian, et al. "Unleashing cheapfakes through trojan plugins of large language models." arXiv preprint arXiv:2312.00374 (2023).
>
> [2] Gu, Naibin, et al. "A gradient control method for backdoor attacks on parameter-efficient tuning." Proceedings of the 61st Annual Meeting of the Association for Computational Linguistics (Volume 1: Long Papers). 2023.
>
> [3] Du, Wei, et al. "PPT: Backdoor Attacks on Pre-trained Models via Poisoned Prompt Tuning." IJCAI. 2022.
>
> ***
>
> >**Finally, we express our sincere gratitude for your review of our paper. We earnestly request that you reconsider our manuscript, and please be assured that we are actively addressing your concerns. If you believe there are areas in need of improvement, we would greatly appreciate your specific feedback on these matters. We are fully available to engage with you until the end of this rebuttal period. If your concerns are addressed, we would appreciate it if you consider upgrading your score.**

---

> ### Author Response · Authors · 2024-11-19
> **Request for Discussion**
>
> Dear Reviewer m5qv:
>
> Kindly note that the author-reviewer discussion period is currently ongoing. We would greatly appreciate it if you could review our response when convenient. We earnestly request that you reconsider our manuscript and consider upgrading your score.
>
> Regards,
>
> Authors

---

> ### Comment · Reviewer_m5qv · 2024-11-21
>
> Thank you for your reply! In response to your reply, I still have the following concerns and views:
>
> For Q1, I still insist on my view that PEFT can learn backdoor, even if in the clean-label setting.
>
> For Q2, I found a defense paper by Zhao et al. that employs the same framework, algorithm, and pipeline as this work. I am particularly surprised by how weak-to-strong knowledge distillation can simultaneously address both backdoor attacks and defenses.
>
> For Q4, the current version is very ambiguous. Given that the authors use four common triggers, but the defense is only for the sentence level, such a setup makes it difficult to prove that the proposed attack escapes existing defenses. Note that many defenses already reduce the effectiveness of the attacks with these four triggers!
>
> In addition, I have the following concerns:
>
> 1. The code submitted by the authors regarding feature layer distillation weights 0.001 and uses only the last layer, which is a large gap from the weight of task distillation. I highly suspect the contribution of the feature layer! Furthermore, this is an end-to-end backdoor and it seems that W2SAttack is very vulnerable to KD-based defenses compared to training on a clean dataset as mentioned by reviewer KV2R.
>
> 2. Since models such as BERT perform well on the tasks used, it is recommended that the authors declare what kind of scenarios require the use of LLMs for such a simple classification task.
>
> 3. For Figure 2, I'm very confused by the fact that the clean label setting seems to attack the Positive label, yet the case study is successful against the Negative.

---

> ### Author Response · Authors · 2024-11-21
> **Further Response to Reviewer m5qv**
>
> Dear Reviewer m5qv,
>
> **Thank you for your reply**.
>
> ***
>
> **Response Q1:** Firstly, in our pilot study, we find that clean-label cannot achieve viable backdoor attack performance in the PEFT setting, **which has been verified through extensive experimentation**. Additionally, we reassert our viewpoint that **clean-label backdoor attacks struggle to establish alignment between the trigger and the target label, but this does not mean that clean label backdoor attacks are completely unfeasible**. For example, in the SynAttack algorithm, the ASR is significantly higher than that of the BadNet algorithm, thereby demonstrating that the success rate of backdoor attacks is also influenced by the form of the trigger.
>
> Furthermore, in Figure 2 of the manuscript, we analyze the impact of different numbers of poisoned samples on the ASR. It is readily apparent that as the number of poisoned samples increases, the ASR gradually improves; however, it remains significantly lower than that achieved through full-parameter fine-tuning.
>
> Therefore, our findings are validated: compared to full-parameter fine-tuning, clean-label backdoor attacks struggle to establish effective alignment between the trigger and the target label in the PEFT setting. We kindly ask you to review our manuscript again and understand the details therein.
>
> ***
>
> **Response Q2:** Thank you for your comments. Although you did not provide the title of the paper, we know which one you are referring to. **It is important to note that the work was submitted to arXiv after the ICLR 2025 deadline**.
>
> We will specifically clarify the differences between the W2SAttack and that work to alleviate your concerns:
>
> The work of Zhao et al. focuses on using knowledge distillation to defend against backdoor attacks [1]. **Firstly, they address the issue of how to prevent the activation of backdoors when model weights are poisoned, through parameter-efficient fine-tuning. Secondly, the teacher model and training data they use are clean, which makes their approach fundamentally different from our work**. Our W2SAttack explores how to enhance the effectiveness of clean-label backdoor attacks in the PEFT setting. We kindly ask you to carefully review the work of Zhao et al. again.
>
> ***
>
> **References:**
>
> [1] Zhao et al. "Unlearning Backdoor Attacks for LLMs with Weak-to-Strong Knowledge Distillation." arXiv preprint arXiv:2410.14425 (2024).

---

> > ### Comment · Reviewer_m5qv · 2024-11-22
> >
> > **Thank you for your reply**
> >
> > **1. For Q1**, I have validated this motivation that the ASR drops when the rank of LoRA is very small. However, the authors assume that W2Attack needs to train a full-parameter small model, e.g. BERT (110M), whereas LoRA's parameter count is typically less than 10M. I would like to see a **Fair Comparison**, i.e., how well LoRA learns the backdoor, tuned for the same parameter count.
> >
> > **2. For Q2**, the author's clarification helped me to understand the difference between the two works, thank you!
> >
> > **However, the author should clarify the issues left over from the last discussion. Also, I'm very confused about the setup at work**
> >
> > 1. I understand W2SAttack to be an **end-to-end backdoor attack**. In other words, the attacker will publish a PEFT to a third-party platform to trick users into adapting local LLMs.So, is the Clean label setting in this article really necessary? Because the attackers have full control over the training process and the poisoned data, using dirty labels will instead reduce the cost of their attack. As the authors said dirty labels are easy to learn backdoor by PEFT. It is well known that the Clean label setup releases a more deceptive dataset and assumes that the user performs fine-tuning without inadvertently learning the backdoor. This seems to contradict the setup of this paper, as users do not seem to employ this type of training to fine-tune their models.
> >
> > 2. Continuing from the previous discussion, the author will release the PEFT module produced by W2SAttack. In the defense evaluation, the authors use only three sample-based inspection methods to demonstrate the robustness of W2SAttack's use of sentence-level triggers, however existing defenses are fully capable of detecting these triggers used in the paper. Thus, the authors over-claim that W2SAttack escapes existing defenses. Furthermore, when publishing a poisoning model, the authors should use a model detection scheme to demonstrate the robustness of W2SAttack.
> >
> > 3. All experimental setups need further clarification and explanation in the manuscript. For example, in Figure 3, it is not clear to me what dataset and triggers the authors used. This also contributes to my poor understanding of the relationship between sample size and poisoning rate.

---

> > > ### Author Response · Authors · 2024-11-22
> > > **Further Response to Reviewer m5qv**
> > >
> > > ***
> > >
> > > **Question 4** All experimental setups need further clarification and explanation in the manuscript. For example, in Figure 3, it is not clear to me what dataset and triggers the authors used. This also contributes to my poor understanding of the relationship between sample size and poisoning rate.
> > >
> > > **Response 4** Figure 3 utilizes the SST-2 dataset and the BadNet backdoor attack algorithm, as do the other figures in the manuscript.
> > >
> > > ***
> > >
> > > **References:**
> > >
> > > [1] Huynh, Tran, et al. "COMBAT: Alternated Training for Effective Clean-Label Backdoor Attacks." Proceedings of the AAAI Conference on Artificial Intelligence. Vol. 38. No. 3. 2024.
> > >
> > > [2] Gao, Yinghua, et al. "Not all samples are born equal: Towards effective clean-label backdoor attacks." Pattern Recognition 139 (2023): 109512.

---

> ### Author Response · Authors · 2024-11-22
> **Further Response to Reviewer m5qv**
>
> **Question Q4** The current version is very ambiguous. Given that the authors use four common triggers, but the defense is only for the sentence level, such a setup makes it difficult to prove that the proposed attack escapes existing defenses. Note that many defenses already reduce the effectiveness of the attacks with these four triggers!
>
> **Response:** Thank you for your comments. **It is worth noting that this paper focuses on backdoor attack algorithms, specifically optimizing the effectiveness of clean-label backdoor attacks in the PEFT setting. The deployment of defense experiments is solely to verify the robustness of the W2SAttack in the face of commonly used defense algorithms. We do not believe that using only sentence-level triggers to test the robustness of the W2SAttack algorithm against defense strategies would lead to ambiguity in the manuscript, including among other reviewers**.
>
> Additionally, whether or not effective defense algorithms exist, this is beyond the scope of our study. Our focus is on enhancing the effectiveness of clean-label backdoor attacks in the PEFT setting.
>
> ***
>
> **Question 1** The code submitted by the authors regarding feature layer distillation weights 0.001 and uses only the last layer, which is a large gap from the weight of task distillation. I highly suspect the contribution of the feature layer! Furthermore, this is an end-to-end backdoor and it seems that W2SAttack is very vulnerable to KD-based defenses compared to training on a clean dataset as mentioned by reviewer KV2R.
>
> **Response:** Thank you for your comments. The configuration of model weights was validated through extensive experiments, and although small, it has a positive effect on enhancing the effectiveness of backdoor attacks. **Based on your comments, should we understand that smaller weights cannot be used in knowledge distillation algorithms**?
>
> Secondly, reviewer KV2R's comment does not pertain to a defense issue, but rather to the verification of whether the W2SAttack algorithm, by using only poisoned teacher model and clean train dataset, can transfer backdoor features in the clean-label setting.
>
> Finally, we reiterate the focus of our work in this paper: we have verified that clean-label backdoor attacks struggle to align triggers with target labels in the same PEFT setting and have introduced the W2SAttack algorithm to enhance the effectiveness of clean-label backdoor attacks. Our core concern is with attack methodologies, not defense.
>
> ***
>
> **Question 2** Since models such as BERT perform well on the tasks used, it is recommended that the authors declare what kind of scenarios require the use of LLMs for such a simple classification task.
>
> **Response:** Thank you for your comments. We understand that BERT has achieved feasible performance in classification tasks, but in many datasets or scenarios, there is still a need for large language models to enhance performance, such as in medical settings where there is a lack of training samples. We have also noted that a significant amount of research utilizes LLMs to accomplish classification tasks [1,2,3].
>
> Additionally, we only use text classification tasks to validate the effectiveness of the W2SAttack algorithm, which is a common strategy in backdoor attacks. There are also numerous studies on backdoor attacks in text classification tasks [4,5,6].
>
> ***
>
> **Question 3** For Figure 2, I'm very confused by the fact that the clean label setting seems to attack the Positive label, yet the case study is successful against the Negative.
>
> **Response:** **We have re-examined Figure 2, and it contains no errors. You seem to not quite understand clean-label backdoor attacks**. The target label for the backdoor attack is 'negative', so it is necessary during the training phase to implant the trigger in samples labeled as 'negative'.
>
> ***
>
> **References:**
>
> [1] Sun, Xiaofei, et al. "Text Classification via Large Language Models." Findings of EMNLP 2023.
>
> [2] Li, Zhuoyan, et al. "Synthetic Data Generation with Large Language Models for Text Classification: Potential and Limitations." The 2023 Conference on Empirical Methods in Natural Language Processing.
>
> [3] Guo, Yuting, et al. "Evaluating large language models for health-related text classification tasks with public social media data." Journal of the American Medical Informatics Association (2024).
>
> [4] You, Wencong, Zayd Hammoudeh, and Daniel Lowd. "Large Language Models Are Better Adversaries: Exploring Generative Clean-Label Backdoor Attacks Against Text Classifiers." The 2023 Conference on Empirical Methods in Natural Language Processing.
>
> [5] Kandpal, Nikhil, et al. "Backdoor Attacks for In-Context Learning with Language Models." The Second Workshop on New Frontiers in Adversarial Machine Learning.
>
> [6] Li, Ziqiang, et al. "Large Language Models are Good Attackers: Efficient and Stealthy Textual Backdoor Attacks." arXiv preprint arXiv:2408.11587 (2024).

---

> ### Author Response · Authors · 2024-11-22
> **Further Response to Reviewer m5qv**
>
> **Question 1:** I have validated this motivation that the ASR drops when the rank of LoRA is very small. However, the authors assume that W2Attack needs to train a full-parameter small model, e.g. BERT (110M), whereas LoRA's parameter count is typically less than 10M. I would like to see a Fair Comparison, i.e., how well LoRA learns the backdoor, tuned for the same parameter count.
>
> **Response 1:** **Thank you for verifying that clean-label backdoor attacks do not achieve effective performance when using LoRA. This confirms that our pilot study was correct. Since this issue has been resolved, could you please update your comment, including Question 1 and its related discussion? This is to avoid any misunderstanding in the meta review**.
>
> In the W2SAttack algorithm, we leverage knowledge distillation as the framework, which necessarily involves the use of a teacher model. Compared to full parameter fine-tuning, the number of parameters that can be updated when using the BERT model is very small. We believe you have reviewed reviewer KV2R's question regarding the communication cost (Question 7) and repeated that issue. In our discussions with reviewer KV2R, we have already resolved his concerns. **We need to correct your error again; the number of parameters in LoRA does not equal 10M, as it varies with the size of the language model**. Additionally, the resources required for fine-tuning the large language model are not solely measured by the number of updatable parameters.
>
> Additionally, employing an extra model to guide the student model is a fundamental requirement of knowledge distillation. We are unclear why the addition of a small-scale teacher model to enhance the effectiveness of backdoor attacks is seen as an unfair comparison.
>
> ***
>
> **Question 2:** I understand W2SAttack to be an end-to-end backdoor attack. In other words, the attacker will publish a PEFT to a third-party platform to trick users into adapting local LLMs.So, is the Clean label setting in this article really necessary? Because the attackers have full control over the training process and the poisoned data, using dirty labels will instead reduce the cost of their attack. As the authors said dirty labels are easy to learn backdoor by PEFT. It is well known that the Clean label setup releases a more deceptive dataset and assumes that the user performs fine-tuning without inadvertently learning the backdoor. This seems to contradict the setup of this paper, as users do not seem to employ this type of training to fine-tune their models.
>
> **Response 2:** We believe this issue completely overlaps with the first question from reviewer LsBW. Thank you for mentioning it again. We will restate our position:
>
> In our pilot study, we found that under the PEFT setting, clean-label backdoor attacks struggle to achieve feasible outcomes compared to full-parameter fine-tuning. To address this issue, this paper introduces a novel backdoor attack algorithm based on feature alignment-enhanced knowledge distillation, aimed at improving the success rate of clean-label backdoor attacks under the PEFT setting. **In summary, our motivation is to enhance the effectiveness of clean-label backdoor attacks; thus, we need to manipulate the training process**.
>
> Secondly, in previous studies, there has been extensive research aimed at optimizing the effectiveness of clean-label backdoor attacks by manipulating the training process [1,2]. In real-world application scenarios, even though attackers do not need to worry about the inspection of training data when they control the training process, we still hope that clean-label backdoor attacks can be more effective.
>
> ***
>
> **Question 3:** Continuing from the previous discussion, the author will release the PEFT module produced by W2SAttack. In the defense evaluation, the authors use only three sample-based inspection methods to demonstrate the robustness of W2SAttack's use of sentence-level triggers, however existing defenses are fully capable of detecting these triggers used in the paper. Thus, the authors over-claim that W2SAttack escapes existing defenses. Furthermore, when publishing a poisoning model, the authors should use a model detection scheme to demonstrate the robustness of W2SAttack.
>
> **Response 3:** The motivation of this paper is to optimize the effectiveness of clean-label backdoor attacks in the PEFT setting; we are not proposing a new form of backdoor attack trigger. Therefore, validating existing methods for detecting triggers does not fall within the scope of this paper.
>
> Additionally, we have tested three common backdoor attack defense methods and found that they are ineffective against models fine-tuned with W2SAttack. We understand that defense efforts are crucial for ensuring model security, which will be the focus of our future research.

---

> ### Comment · Reviewer_m5qv · 2024-11-23
>
> **Thank you for your reply!**
>
> **For Q1:** I don't still understand this claim "but rather to the verification of whether the...".  Moreover, this can exist as a fair comparison. For example, using the same parameters and computation resource in a clean label setup, i.e., increasing the rank of LoRA to achieve full parametric fine-tuning of BERT, can strengthen the motivation of this paper!
>
> **For Q2:** With the W2SAttack motivation established, I suggest that the authors correct the attack scenarios in this paper. As the reviewer LsBW worries, the clean label simply releases a more hidden dataset, which leads to a backdoor trained by the user. Therefore, I still disagree that an attacker needs to set a clean label to publish a backdoor model. Recently, Weak to Strong has been equally effective in terms of task performance. Perhaps, when the user adopts that approach to train the model under the clean label dataset released by the attacker may generate a W2SAttack. this should be a realistic attack scenario!
>
> **For Q3:** On the left side of Figure 2, it is observed that the trigger is inserted in the positive label of the sentence, but on the right side, it is inserted in the negative.
>
> **For Q4:** It is clear that W2SAttack is an effective attack strategy under the clean label setting. However, the defense experiments regarding the trigger settings and purpose need to be clarified in the revised manuscript. Since the focus of W2SAttack is not on trigger design, it should not claim to be able to escape existing defense strategies. The authors should theoretically and experimentally analyze the possible defenses to be encountered in a realistic attack scenario.
>
> Moreover, I still look forward to seeing detailed clarifications from the authors in the revised edition on experimental setups, etc., which will help to understand more about how the assessment was performed. Of course, I would be open to a more in-depth discussion on W2SAttack. For now, some of my concerns have been better addressed and I will raise my rating appropriately based on the revised version!

---

> > ### Author Response · Authors · 2024-11-24
> > **Further Response to Reviewer m5qv**
> >
> > **Question 5** Moreover, I still look forward to seeing detailed clarifications from the authors in the revised edition on experimental setups, etc., which will help to understand more about how the assessment was performed.
> >
> > **Response 5:** To alleviate your concerns, we have detailed descriptions of the poisoned dataset, victim model, and backdoor attack algorithm in every Table and Figure that contains experimental assessments.
> >
> > ***
> >
> > **References:**
> >
> > [1] Zhou, Zhanhui, et al. "Weak-to-Strong Search: Align Large Language Models via Searching over Small Language Models." NeurIPS 2024.
> >
> > [2] Burns, Collin, et al. "Weak-to-Strong Generalization: Eliciting Strong Capabilities With Weak Supervision." Forty-first International Conference on Machine Learning.
> >
> > [3] Zhao, Xuandong, et al. "Weak-to-strong jailbreaking on large language models." arXiv preprint arXiv:2401.17256 (2024).

---

> > ### Author Response · Authors · 2024-11-25
> > **Request for Feedback on Rebuttal**
> >
> > Dear Reviewer m5qv:
> >
> > Kindly note that the author-reviewer discussion period is ending. We would greatly appreciate it if you could review our response at your convenience. We earnestly request that you reconsider our manuscript and consider upgrading your score.
> >
> > Regards,
> >
> > Authors

---

> > > ### Comment · Reviewer_m5qv · 2024-11-26
> > >
> > > Dear Authors,
> > >
> > > Following our discussion, I believe the paper has seen significant improvements. However, as noted by reviewer LsBW, releasing a poisoned model appears more realistic. Even in the clean label setting, the attacker need only adjust the inner rank of LoRA (e.g., 512).
> > >
> > > In the context of Weak-to-Strong (W2S) user scenarios, this threat may appear less significant, as not all users would need to adopt this approach to enhance model performance. Considering that W2S attacks are less impactful and more difficult to implement than dirty labeling or adjusting the inner rank of LoRA, I believe this attack is still far from meeting the standards required for the ICLR conference.
> > >
> > > I would like to thank the authors again for their explanations during this review period, and I will maintain my current rating.
> > >
> > > Best regards,
> > >
> > > Reviewer m5qv

---

> ### Author Response · Authors · 2024-11-24
> **Further Response to Reviewer m5qv**
>
> Dear Reviewer m5qv,
>
> **Thank you for your reply**.
>
> **Question 1** I don't still understand this claim "but rather to the verification of whether the...". Moreover, this can exist as a fair comparison. For example, using the same parameters and computation resource in a clean label setup, i.e., increasing the rank of LoRA to achieve full parametric fine-tuning of BERT, can strengthen the motivation of this paper!
>
> **Response 1:** Thank you for your comments. What we intend to express is that in the comment by reviewer KV2R, they suggest that we include an experiment that solely utilizes the poisoned teacher model for comparison, meaning the training samples are clean; we have added the relevant experiment accordingly.
>
> Your recommendation to conduct experiments with an increased rank under the same conditions is indeed valuable. **In fact, on page 19 of the manuscript in Figure 6, we have already conducted the relevant experiment. We suggest you review our manuscript again**. The results are presented in Table 1.
>
> | Rank|8|   | 32 |  | 64 |    | 128 |  |
> |:---------:|:------:|:------:|:-------:|:-----:|:---:|:--:|:---:|:--:|
> |Metrics| CA | ASR | CA | ASR | CA | ASR | CA | ASR |
> |Results | 95%|15.51% | 95.99%| 40.04%| 94.84% | 41.03% |95.44% | 59.63% |
> |
>
> Table 1: The impact of the number of updatable parameters on ASR.
>
> ***
>
> **Question 2** With the W2SAttack motivation established, I suggest that the authors correct the attack scenarios in this paper. As the reviewer LsBW worries, the clean label simply releases a more hidden dataset, which leads to a backdoor trained by the user. Therefore, I still disagree that an attacker needs to set a clean label to publish a backdoor model. Recently, Weak to Strong has been equally effective in terms of task performance. Perhaps, when the user adopts that approach to train the model under the clean label dataset released by the attacker may generate a W2SAttack. this should be a realistic attack scenario!
>
> **Response 2:** Thank you for your comment. Descriptions of the application scenarios have already been added to the manuscript. On page 7 of the manuscript:
>
> > The potential applications of W2SAttack may be utilized in weak-to-strong model scenarios [1,2,3], which leverage small-scale models to enhance the performance of LLMs.
>
> On page 21 of the manuscript:
>
> > Existing research indicates that leveraging small-scale language models as guides has the potential to enhance the performance of LLMs. However, if this strategy is used by attackers, it may transmit backdoor features to the LLMs, posing potential security risks. Therefore, the potential applications of W2SAttack may be utilized in weak-to-strong model scenarios, which involve poisoning LLMs in the clean-label setting.
>
> ***
>
> **Question 3** On the left side of Figure 2, it is observed that the trigger is inserted in the positive label of the sentence, but on the right side, it is inserted in the negative.
>
> **Response 3:** Thank you for your comments. **We have checked Figure 2 repeatedly; it contains no errors, perhaps you do not understand clean-label backdoor attacks**. **In Figure 2, we insert the trigger "mn" into the sentence "The road is muddy," which is a negative sample, not a positive one**. Additionally, on the right side of Figure 2, the model inference, which implants the trigger into the positive sample but classifies it as negative, indicates that the backdoor attack has been successful.
>
> ***
>
> **Question 4** It is clear that W2SAttack is an effective attack strategy under the clean label setting. However, the defense experiments regarding the trigger settings and purpose need to be clarified in the revised manuscript. Since the focus of W2SAttack is not on trigger design, it should not claim to be able to escape existing defense strategies. The authors should theoretically and experimentally analyze the possible defenses to be encountered in a realistic attack scenario.
>
> **Response 4:** Thank you for your comments. **We have revised the description of the defense experiment analysis, which includes using the sentence "I watched this 3D movie" as a trigger, and we state that W2SAttack demonstrates stability against the existing three defense methods, rather than all methods**. Additionally, based on your suggestion, we are considering a potential defense strategy that involves using an ensemble of multiple small-scale teacher models to construct a mixture-of-teachers model. This model collaboratively guides the LLMs, thereby preventing the transmission of backdoors.

---

> ### Author Response · Authors · 2024-11-26
> **Response to Reviewer m5qv**
>
> Dear Reviewer m5qv,
>
> **Thank you for your reply**.
>
> ***
>
> **Response :** First, thank you for acknowledging the improvements in the quality of our paper.
>
> We need to clarify that if we were only to consider the ASR, most existing backdoor attack algorithms could achieve a 100% success rate. As members of the LLM community, our motivation for designing backdoor attack algorithms is to explore potential security vulnerabilities. It is evident that a feasible ASR can be achieved by adjusting the amount of updatable parameters. But just because it is possible to achieve a 100% ASR by adjusting the rank, should we abandon research into other backdoor attack algorithms?
>
> Although scenarios involving Weak-to-Strong are not very common, we believe that to build a reliable LLM community, it is necessary to consider such potential security issues. We cannot afford to ignore these security risks simply because they are rare. We have consistently responded according to your concerns and have received your affirmation. The current refusal to modify the score, based on the argument that the algorithm's application scenario is niche, is difficult to accept.

---

### Official Review · Reviewer_KV2R · 2024-10-24

**Soundness:** 3
**Presentation:** 3
**Contribution:** 3
**Rating:** 8
**Confidence:** 4

**Summary:**

The preliminary experiments in this paper discovered that the PEFT, which updates only a small number of model parameters, hardly implements backdoor attacks effectively. Based on these findings, the authors proposed a weak-to-strong backdoor attack algorithm targeting PEFT, named W2SAttack. They leverage a small-scale teacher model to facilitate the student model's learning of backdoor features, thereby enhancing the effectiveness of the backdoor attack.

**Strengths:**

1.	Enhancing the effectiveness of backdoor attacks targeting the PEFT algorithm is a worthwhile research problem.
2.	The authors design an effective backdoor attack algorithm that saves computational resources compared to full-parameter fine-tuning.
3.	Overall, the presentation is clear, and the experiments are comprehensive. The details are clear.

**Weaknesses:**

Some aspects are not clear, see the questions section.

**Questions:**

More Explanation:
	Figure 1's y-axis should have a label; I was confused about its unit and what it represents.
	CA and ASR should be clearly mentioned in the caption.
	During the process of poisoning the teacher model, the authors added an additional linear layer. Is this layer necessary? Equation 4 requires further modification. What are the impacts of the teacher model on backdoor attacks?
	The expression of the attacker's Objective 1 indeed requires additional explanation. The authors have noted in their third stage pilot study that deploying effective backdoor attacks using the PEFT algorithm is challenging. However, Objective 1 suggests that ASR(f(x^' )_peft)≈ASR(f(x^' )_fpft), a statement that seems to contradict the earlier assessment, which requires further explanation.
Experimental Section:
	Is the caption for Figure 4 correct? The authors discuss the impact of different trigger lengths on backdoor attacks in the experimental analysis section, therefore this part needs to be revised.
	Compared to the BadNets backdoor attack, the backdoor attack algorithms based on InSent or SynAttack seem to achieve more desirable effects. Could the authors provide further analysis of the reasons for this, or present a more detailed analysis?
	Although the W2SAttack algorithm can guide the model to learn backdoor features, it requires the design of an additional teacher model. Existing experiments have only analyzed the effectiveness of the backdoor attack, but lack necessary analyses of communication costs, such as the training costs induced by changes in updatable parameters, which are essential for assessing the feasibility of the algorithm.

---

> ### Author Response · Authors · 2024-11-16
> **Response to Reviewer KV2R**
>
> Dear Reviewer KV2R,
>
> **Thank you for your review!** We have endeavored to address all your questions and concerns below. Please let us know if there are any aspects that we need to sufficiently clarify. **If you feel that your concerns have been satisfactorily addressed, we would be grateful if you would consider revising your score.** Please do not hesitate to reach out with any further questions. We value your feedback and welcome any additional queries.
>
> ***
>
> **Question 1:** Figure 1's y-axis should have a label; I was confused about its unit and what it represents.
>
> **Response 1:** Thank you for your comments. In Figure 1, we present the experimental results of the pilot study. This figure compares the clean accuracy and the success rate of backdoor attacks under different fine-tuning settings. We have revised Figure 1, and thank you for your suggestions.
>
> ***
>
> **Question 2:** CA and ASR should be clearly mentioned in the caption.
>
> **Response 2:** Thank you for your comments. CA and ASR represent the clean accuracy and the attack success rate of the poisoned large language model on clean and poisoned samples, respectively, which are common evaluation metrics for backdoor attacks. Thank you for your suggestions; we have supplemented the caption. For a more detailed introduction to the evaluation metrics, please refer to the Appendix B.
>
> ***
>
> **Question 3:** During the process of poisoning the teacher model, the authors added an additional linear layer. Is this layer necessary? Equation 4 requires further modification. What are the impacts of the teacher model on backdoor attacks?
>
> **Response 3:** Thank you for your comments. In the W2SAttack, to further enhance the transfer of backdoor features, we need to calculate the feature alignment loss between the small-scale teacher model and the large-scale student model based on the final hidden states. **For example, the size of the final hidden states of the BERT model is 768, while that of the LLaMA model is 4096. Therefore, it is necessary for the W2SAttack algorithm to add an additional linear layer to ensure the alignment of feature dimensions between the teacher model and the student model**. We have made further modifications to Equation 4 to clarify its role.
>
> In the W2SAttack, the role of the teacher model is to transfer backdoor features to the large-scale student model through feature alignment-enhanced knowledge distillation, enabling the student model to effectively implement a backdoor attack while updating only a minimal number of model parameters. Thank you once again for your comments.
>
> ***
>
> **Question 4:** The expression of the attacker's Objective 1 indeed requires additional explanation. The authors have noted in their third stage pilot study that deploying effective backdoor attacks using the PEFT algorithm is challenging. However, Objective 1 suggests that ASR(f(x^' )_peft)≈ASR(f(x^' )_fpft), a statement that seems to contradict the earlier assessment, which requires further explanation.
>
> **Response 4:** Thank you for your comments. As attackers, the motivation behind Objective 1 is to develop a new backdoor attack algorithm that achieves success rates comparable to those of full-parameter fine-tuning while utilizing parameter-efficient fine-tuning. **Therefore, $ASR(f(x')_{peft})$ represents the attack success rate after using the W2SAttack algorithm**. Thank you for your suggestions; we have revised this statement in the manuscript.
>
> ***
>
> **Question 5:** Is the caption for Figure 4 correct? The authors discuss the impact of different trigger lengths on backdoor attacks in the experimental analysis section, therefore this part needs to be revised.
>
> **Response 5:** Thank you for your suggestions. The motivation behind Figure 4 is to demonstrate the impact of different trigger lengths on backdoor attacks in settings of full-parameter fine-tuning and parameter-efficient fine-tuning. We have revised the caption of Figure 4 in the manuscript accordingly.

---

> ### Author Response · Authors · 2024-11-16
> **Response to Reviewer KV2R**
>
> **Question 6:** Compared to the BadNets backdoor attack, the backdoor attack algorithms based on InSent or SynAttack seem to achieve more desirable effects. Could the authors provide further analysis of the reasons for this, or present a more detailed analysis?
>
> **Response 6:** Thank you for your comments. Compared to the BadNet backdoor attack algorithm, the trigger features used by InSent or SynAttack are more distinct. For instance, in the InSent algorithm, we use "I watched this 3D movie" as the trigger, which is more conspicuous than the rare character trigger "mn" used by BadNet. Consequently, it is easier to establish an alignment between the trigger and the target label, resulting in a higher success rate for the backdoor attack. Similarly, in the SynAttack algorithm, we use "(SBARQ (WHADVP) (SQ) (.))" as an abstract syntactic trigger, which is more distinct. Therefore, even when using parameter-efficient fine-tuning, the success rate of the backdoor attack remains high.
>
> ***
>
> **Question 7:** Although the W2SAttack algorithm can guide the model to learn backdoor features, it requires the design of an additional teacher model. Existing experiments have only analyzed the effectiveness of the backdoor attack, but lack necessary analyses of communication costs, such as the training costs induced by changes in updatable parameters, which are essential for assessing the feasibility of the algorithm.
>
> **Response 7:** Thank you for your comments. In the W2SAttack algorithm, to enhance the success rate of backdoor attacks in parameter-efficient fine-tuning settings, we introduce a small-scale teacher model to guide the student model in establishing alignment between the trigger and the target label. Compared to implementing the backdoor attack with full-parameter fine-tuning on large language models, the W2SAttack algorithm requires significantly fewer computational resources. As shown in Table 1, in the OPT model, the number of parameters fine-tuned with full-parameter tuning is 1,317,339,136, whereas the W2SAttack algorithm only needs to update a total of 1,576,960 parameters. Additionally, the computational resources required for the W2SAttack algorithm amount to only 8.55% of those needed for full-parameter fine-tuning, including 111,062,788 parameters of the teacher model. Therefore, the W2SAttack algorithm not only achieves feasible attack performance but also demonstrates good efficiency. Thank you for your suggestions; we include related experimental analysis in the manuscript.
>
> |       |FPFT      | W2SAttack |Ratio   |
> |:---------:|:-------------:|:-------------:|:----------:|
> | Parameters | 1,317,339,136| 112,639,748| 8.55%|
> |
>
> Table 1: The comparison of parameters between full-parameter fine-tuning and the W2SAttack algorithm.
>
> ***
>
> >**In the end, we express our sincere gratitude for your detailed comments, which have been instrumental in improving our work. We greatly appreciate your insights and look forward to further discussions during the rebuttal. If there are any additional questions or concerns, please do not hesitate to share them with us. We remain at your disposal until the end of this rebuttal period.**

---

> > ### Comment · Reviewer_KV2R · 2024-11-19
> > **Response to Authors**
> >
> > I have read the author's response, thank you for the reply! All my concerns mentioned above have been addressed. Additionally, I have some further questions after reviewing the manuscript:
> >
> > 1. The manuscript has moved part of the Related Work section from the appendix to the main body, which facilitates the reader's understanding of the W2SAttack algorithm. However, the sentence 'In this section, we introduce work related to this study, which includes backdoor attacks, parameter-efficient fine-tuning algorithms, and knowledge distillation.' needs to be modified, as the work on knowledge distillation is included in the main body.
> >
> > 2. Some work related to backdoor attacks based on knowledge distillation needs to be discussed. Such as: A knowledge distillation-based backdoor attack in federated learning; Revisiting Data-Free Knowledge Distillation with Poisoned Teachers
> >
> > 3. The author needs to further discuss if, when the training data for the large-scale student model is clean, the backdoor can still be effectively transferred and implemented using only a poisoned teacher model.

---

> > > ### Author Response · Authors · 2024-11-21
> > > **Further Response to Reviewer KV2R**
> > >
> > > Dear Reviewer KV2R,
> > >
> > > **Thank you for reviewing our paper again!** We have endeavored to address all your questions and concerns below. Please let us know if there are any aspects that we need to sufficiently clarify. **If you feel that your concerns have been satisfactorily addressed, we would be grateful if you would consider revising your score.** Please do not hesitate to reach out with any further questions. We value your feedback and welcome any additional queries.
> > >
> > > ***
> > >
> > > **Question 1:** The manuscript has moved part of the Related Work section from the appendix to the main body, which facilitates the reader's understanding of the W2SAttack algorithm. However, the sentence 'In this section, we introduce work related to this study, which includes backdoor attacks, parameter-efficient fine-tuning algorithms, and knowledge distillation.' needs to be modified, as the work on knowledge distillation is included in the main body.
> > >
> > > **Response 1:** Thank you for your recognition and thorough reading. In the latest manuscript, to facilitate understanding of the W2SAttack algorithm, we have moved part of the related work to the main body. Therefore, the description of Related Work in the appendix requires modification. Thank you for your suggestion. The revision is as follows:
> > >
> > > >In this section, we introduce additional work related to this study, which includes backdoor attacks and parameter-efficient fine-tuning algorithms.
> > >
> > > ***
> > >
> > > **Question 2:** Some work related to backdoor attacks based on knowledge distillation needs to be discussed. Such as: A knowledge distillation-based backdoor attack in federated learning; Revisiting Data-Free Knowledge Distillation with Poisoned Teachers
> > >
> > > **Response 2:** Thank you for your comments. We have added a discussion of the literature to the manuscript:
> > >
> > > >Wang et al. [1] leverage knowledge distillation to reduce anomalous features in model outputs caused by label flipping, enabling the model to bypass defenses and increase the attack success rate. Hong et al. [2] uncover that backdoors can be transferred from the poisoned teacher model to the student model in the data-free knowledge distillation setting.
> > >
> > > ***
> > >
> > > **Question 3:** The author needs to further discuss if, when the training data for the large-scale student model is clean, the backdoor can still be effectively transferred and implemented using only a poisoned teacher model.
> > >
> > > **Response 3:** Thank you for your comments. To further alleviate your concerns, we explored the performance of backdoor attacks using only poisoned teacher models, while the training data for the large-scale student model remains clean. **The experimental results are shown in Table 1. It is not difficult to observe that using only a poisoned teacher model cannot effectively transfer backdoors. This is due to the fact that, under the PEFT setting, clean label backdoor attacks struggle to establish feature alignment between triggers and target labels, failing to achieve the viable attack success rates**.
> > >
> > > |       |SST-2|            | CR     |       | AG’s News |    |
> > > |:---------:|:-------------:|:---------:|:----------:|:---------:|:----------:|:---------:|
> > > |Method| CA | ASR | CA | ASR | CA | ASR |
> > > |W2SAttack | 95.55|99.45 | 90.58| 97.71 | 91.79 | 97.39 |
> > > |Clean_data | 95.94|2.42  | 89.55| 1.87  | 91.74 | 2.21 |
> > > |
> > >
> > > Table 1: The results of our W2SAttack algorithm in PEFT. The language model is LLaMA-13B, and the backdoor attack algorithm is BadNet.
> > >
> > > ***
> > >
> > > **References:**
> > >
> > > [1] Wang, Yifan, et al. "A knowledge distillation-based backdoor attack in federated learning." arXiv preprint arXiv:2208.06176 (2022).
> > >
> > > [2] Hong, Junyuan, et al. "Revisiting data-free knowledge distillation with poisoned teachers." International Conference on Machine Learning. PMLR, 2023.
> > >
> > > ***
> > >
> > > >**We sincerely thank you for your detailed comments, which have greatly improved our work. We appreciate your insights and look forward to discussing them further during the rebuttal period. Please feel free to share any additional questions or concerns. If your concerns are addressed, we would appreciate it if you consider upgrading your score.**

---

> > > > ### Author Response · Authors · 2024-11-26
> > > > **Request for Feedback on Rebuttal**
> > > >
> > > > Dear Reviewer KV2R:
> > > >
> > > > Kindly note that the author-reviewer discussion period is ending. We would greatly appreciate it if you could review our response at your convenience. We earnestly request that you reconsider our manuscript and consider upgrading your score.
> > > >
> > > > Regards,
> > > >
> > > > Authors

---

> > > > > ### Comment · Reviewer_KV2R · 2024-11-26
> > > > > **Response to Authors**
> > > > >
> > > > > Thank you for the response. I have read the author's reply and the revised manuscript, and my concerns have been addressed. Furthermore, Reviewer m5qv has validated the pilot experiments of the manuscript, but I encourage the authors to explore more factors that influence the success of backdoor attacks under the PEFT setting, which could be enlightening for subsequent research. I will also consider revising my score.

---

> > > > > > ### Author Response · Authors · 2024-11-26
> > > > > > **Response to Reviewer KV2R**
> > > > > >
> > > > > > Dear Reviewer KV2R,
> > > > > >
> > > > > > **Thank you for your review!**
> > > > > >
> > > > > > ***
> > > > > >
> > > > > > **Question :** Furthermore, Reviewer m5qv has validated the pilot experiments of the manuscript, but I encourage the authors to explore more factors that influence the success of backdoor attacks under the PEFT setting, which could be enlightening for subsequent research.
> > > > > >
> > > > > > **Response :** Thank you for your recognition and thorough reading. **Following your suggestions, we analyzed several factors that influence the attack success rate, including the number of poisoned samples, the number of parameters that can be updated in the model, and the model architecture**.
> > > > > >
> > > > > > As shown in Table 1, the attack success rates gradually increases with the number of poisoned samples. However, too many poisoned samples may increase the risk of exposing the backdoor.
> > > > > >
> > > > > > | Number|250|   | 500 |  | 750 |    | 1000 |  | 1250 |    | 1500 |    |
> > > > > > |:---------:|:------:|:------:|:-------:|:-----:|:---:|:--:|:---:|:--:|:---:|:--:|:---:|:--:|
> > > > > > |Metrics| CA | ASR | CA | ASR | CA | ASR | CA | ASR | CA | ASR | CA | ASR |
> > > > > > |Results | 96.05%|4.73% | 95.83%| 6.49%| 95.77% | 9.02% |95% | 15.51% |94.67% | 47.41% |95.11% | 54.57% |
> > > > > > |
> > > > > >
> > > > > > Table 1: The impact of the number of poisoned samples on ASR.
> > > > > >
> > > > > > As shown in Table 2, the attack success rates continuously improves as the number of updatable parameters in the model increases.
> > > > > >
> > > > > > | Rank|8|   | 32 |  | 64 |    | 128 |  |
> > > > > > |:---------:|:------:|:------:|:-------:|:-----:|:---:|:--:|:---:|:--:|
> > > > > > |Metrics| CA | ASR | CA | ASR | CA | ASR | CA | ASR |
> > > > > > |Results | 95%|15.51% | 95.99%| 40.04%| 94.84% | 41.03% |95.44% | 59.63% |
> > > > > > |
> > > > > >
> > > > > > Table 2: The impact of the number of updatable parameters on ASR.
> > > > > >
> > > > > > As shown in Table 3, we analyzed the impact of different network architectures on the ASR, which includes encoder-decoder (BERT) and decoder-only (OPT) models. It is evident that the attack success rates are relatively low for both types of models. Furthermore, in Table 1 of the manuscript, we find that different forms of triggers also have an impact on the ASR. For example, sentence-level triggers significantly outperform character-level triggers.
> > > > > >
> > > > > > | Model | BERT|   |  OPT |  |
> > > > > > |:---------:|:------:|:------:|:-------:|:-----:|
> > > > > > |Metrics| CA | ASR | CA | ASR |
> > > > > > |Results | 87.75%|19.25% | 95%|15.51%|
> > > > > > |
> > > > > >
> > > > > > Table 3: The impact of different model architectures on ASR.
> > > > > >
> > > > > > ***
> > > > > >
> > > > > > **Thank you again for your suggestions and assistance. If your concerns have been addressed, we would appreciate it if you would consider upgrading your score. Please let us know if you have any further questions. We are actively available until the end of this rebuttal period**.

---

> > > > > > > ### Author Response · Authors · 2024-11-28
> > > > > > > **Request for Feedback on Rebuttal**
> > > > > > >
> > > > > > > Dear Reviewer KV2R:
> > > > > > >
> > > > > > > We thank you for the valuable comments. The discussion period is nearing its conclusion. Please let us know if you have any further concerns. We are looking forward to your feedback.
> > > > > > >
> > > > > > > Regards,
> > > > > > >
> > > > > > > Authors

---

> > > > > > > > ### Comment · Reviewer_KV2R · 2024-12-01
> > > > > > > >
> > > > > > > > I would like to thank the authors for their response and efforts in the rebuttal. I have read the revised manuscript and the responses to other reviewers. I believe my concerns have been addressed, and I decided to upgrade both my score and confidence. I recommend accepting this paper, good work.

---

> > > > > > > > > ### Author Response · Authors · 2024-12-02
> > > > > > > > > **Thank you for increasing your score!**
> > > > > > > > >
> > > > > > > > > Thank you for upgrading your score! We appreciate the time and effort you dedicated to reviewing this work.

---

> ### Author Response · Authors · 2024-11-19
> **Request for Discussion**
>
> Dear Reviewer KV2R:
>
> Kindly note that the author-reviewer discussion period is currently ongoing. We would greatly appreciate it if you could review our response when convenient. We earnestly request that you reconsider our manuscript and consider upgrading your score.
>
> Regards,
>
> Authors

---

### Official Review · Reviewer_uUxZ · 2024-10-24

**Soundness:** 3
**Presentation:** 3
**Contribution:** 3
**Rating:** 8
**Confidence:** 3

**Summary:**

This paper proposed a method called W2SAttack. The author claims 1. that full-parameter fine-tune for achieving backdoor attack is not feasible due to high occupied VRAM 2. PEFT such as LoRA causes poor performance.
To address the posed problem, the author proposed the W2SAttack. They use the PEFT for smaller LLM first, and then set it as the teacher model to distill the larger LLM.
The results showed that the method can significantly reduce the computational cost.

**Strengths:**

1.	The article proposes a counter-intuitive but effective framework, that is, using small models as teachers and large models as students. This makes me think it's quite novel
2.	The writing is fluent and clear, easy to understand

**Weaknesses:**

Some weaknesses can be found in the Questions.

**Questions:**

The definition of ASR(f(x^' )_peft) in Obj. 1 needs further clarification.
	What is the definition of Z_t? Additionally, the author needs to further explain why I(Z_t;Y) is related to backdoor features.
	I didn’t find the implementation details for Eq. 9 and 10, particularly for Eq. 9. Thus I have a concern about their correctness. Please provide more details.
	The author said they use the clean-label backdoor attack. Why don’t use the poison-label backdoor attack? Is there any difference between those two attacks in your method? Please clarify. Besides, the author should provide the details for attack, such as target label to solve my concern.
	I wonder if continuously increasing the number of poisoned samples would improve the attack success rate in the PEFT setting?
	The method of inserting triggers also affects the attack success rates. The author needs to further explain the implementation details of BadNet and InSent, as well as the SynAttack algorithm.
	The caption for Fig. 3 should provide a detailed description of the motivation for each subfigure.
	What is the meaning of ‘Efficient-tuning’ in Tab. 5?
	Also a concern about reproducibility, in Tab. 9, there is few details provided in terms of defense. For example, which trigger you used?

---

> ### Author Response · Authors · 2024-11-16
> **Response to Reviewer uUxZ**
>
> Dear Reviewer uUxZ,
>
> **Thank you for your review!** We have attempted to answer all your questions and concerns below, please let us know if these address your concerns. **If you feel that your concerns have been satisfactorily addressed, we would be grateful if you would consider revising your score.** Please do not hesitate to reach out with any further questions. We value your feedback and welcome any additional queries.
>
> ***
>
> **Question 1:** The definition of $ASR(f(x')_{peft})$ in Obj. 1 needs further clarification.
>
> **Response 1:** Thank you for your comments. The motivation behind Objective 1 is to demonstrate that after leveraging our proposed feature alignment-enhanced knowledge distillation algorithm, the model's backdoor attack success rate approaches that of full-parameter fine-tuning. **Therefore, $ASR(f(x')_{peft})$ represents the attack success rate after the use of the W2SAttack algorithm**. Thank you for your suggestions; we revise the definition of Objective 1 in the manuscript.
>
> ***
>
> **Question 2:** What is the definition of $Z_t$? Additionally, the author needs to further explain why $I(Z_t; Y)$ is related to backdoor features.
>
> **Response 2:** Thank you for your comments. $Z_t$ represents the intermediate features of the poisoned teacher model. In the W2SAttack algorithm, the small-scale teacher model utilizes full-parameter fine-tuning to establish alignment between the trigger and the target label; therefore, the mutual information $I(Z_t; Y)$ is related to backdoor features. Thank you for your suggestions; we include an explanation of $I(Z_t; Y)$ in the manuscript.
>
> ***
>
> **Question 3:** I didn’t find the implementation details for Eq. 9 and 10, particularly for Eq. 9. Thus I have a concern about their correctness. Please provide more details.
>
> **Response 3:** Thank you for your comments. In Equations 9 and 10, our motivation is to facilitate the alignment of backdoor features between the poisoned teacher model and the student model. Specifically, we obtain the final hidden states of the teacher and student models, calculate their Euclidean distance, and optimize the feature alignment loss.
>
> Thank you for your suggestions; we have revised the equations accordingly.
>
> ***
>
> **Question 4:** The author said they use the clean-label backdoor attack. Why don’t use the poison-label backdoor attack? Is there any difference between those two attacks in your method? Please clarify. Besides, the author should provide the details for attack, such as target label to solve my concern.
>
> **Response 4:** Thank you for your comments. **The reason for choosing the clean-label backdoor attack lies in our pilot study, where we verify that when fine-tuning language models with the PEFT algorithm, clean-label backdoor attacks may not establish an effective alignment between the trigger and the target label**. To address this issue, we propose the W2SAttack algorithm.
>
> Compared to clean-label backdoor attacks, poison-label backdoor attacks require the attacker to modify the labels of the poisoned samples, thereby establishing association between the trigger and the target output. This association helps the model learn backdoor features more effectively. Previous research has shown that clean-label backdoor attacks require more poisoned samples compared to poison-label backdoor attacks [1]. Similarly, in our experiments, we also observe that as the number of poisoned samples increases, the success rate of backdoor attacks gradually increases. Therefore, the purpose of this paper is to explore how to fine-tune the student model with the fewest poisoned samples under the clean-label backdoor attack setting to achieve the optimal backdoor attack success rate.
>
> Thank you for your comments. Regarding the description of the backdoor attack target label, on page 18 of the manuscript, the target labels are selected as "negative," "negative," and "world" for different datasets.
>
> ***
>
> **Question 5:** I wonder if continuously increasing the number of poisoned samples would improve the attack success rate in the PEFT setting?
>
> **Response 5:** Thank you for your comments. Your assumption is correct. As shown in Figure 3 of the manuscript, the attack success rate gradually increases as the number of poisoned samples increases. Although increasing the number of poisoned samples can enhance the success rate of backdoor attacks, an excessive number of poisoned samples also raises the risk of the attack being detected by defense algorithms. Therefore, the motivation of this paper is to explore how to poison large language models using a minimal number of poisoned samples, under the premise of efficient parameter fine-tuning.

---

> ### Author Response · Authors · 2024-11-16
> **Response to Reviewer uUxZ**
>
> **Question 6:** The method of inserting triggers also affects the attack success rates. The author needs to further explain the implementation details of BadNet and InSent, as well as the SynAttack algorithm?
>
> **Response 6:** Thank you for your comments. For the BadNet and InSent attacks, we respectively choose the rare character "mn" and the sentence "I watched this 3D movie" as triggers, which we randomly insert into the training samples. For the SynAttack algorithm, we select "( SBARQ ( WHADVP ) ( SQ ) ( . ) )" as an abstract syntactic trigger. For further details, please refer to Appendix B.
>
> ***
>
> **Question 7:** The caption for Fig. 3 should provide a detailed description of the motivation for each subfigure.
>
> **Response 7:** Thank you for your comments. In Figure 3, we analyze the impact of different numbers of poisoned samples on the success rate of backdoor attacks under full-parameter fine-tuning and parameter-efficient fine-tuning settings. It is evident that as the number of poisoned samples increases, the success rate of backdoor attacks gradually increases under the parameter-efficient fine-tuning setting. Thank you for your suggestion; we have added a detailed caption to Figure 3.
>
> ***
>
> **Question 8:** What is the meaning of 'Efficient-tuning' in Tab. 5?
>
> **Response 8:** In Table 5, we aim to demonstrate the effectiveness of the W2SAttack algorithm under different fine-tuning strategies. "Efficient-tuning" refers to the parameter-efficient fine-tuning algorithms. Thank you for your comments; we have provided further explanations in the manuscript.
>
> ***
>
> **Question 9:** Also a concern about reproducibility, in Tab. 9, there is few details provided in terms of defense. For example, which trigger you used?
>
> **Response 9:** To verify the stability of the W2SAttack algorithm against defense mechanisms, we deployed commonly used defense algorithms. During the attack phase, we used InSent as the backdoor attack algorithm, with the sentence "I watched this 3D movie" as the trigger. Thank you for your comments; we have added an introduction to the triggers in this Table.
>
> ***
>
> **References:**
>
> [1] Barni, Mauro, Kassem Kallas, and Benedetta Tondi. "A new backdoor attack in cnns by training set corruption without label poisoning." 2019 IEEE International Conference on Image Processing (ICIP). IEEE, 2019.
>
> ***
>
> >**In the end, thanks a lot for your detailed comments and thank you for helping us improve our work! We appreciate your thoughts on our work and we would be more than happy to discuss more during the rebuttal. If your concerns are addressed, we would appreciate it if you considered upgrading your score. Please let us know if you have any further questions. We are actively available until the end of this rebuttal period.**

---

> ### Author Response · Authors · 2024-11-19
> **Request for Discussion**
>
> Dear Reviewer uUxZ:
>
> Kindly note that the author-reviewer discussion period is currently ongoing. We would greatly appreciate it if you could review our response when convenient. We earnestly request that you reconsider our manuscript and consider upgrading your score.
>
> Regards,
>
> Authors

---

> > ### Comment · Reviewer_uUxZ · 2024-11-24
> > **Response to Authors**
> >
> > I have read the reviews and responses from the author and am positively inclined towards accepting the paper.

---

### Official Review · Reviewer_LsBW · 2024-11-03

**Soundness:** 2
**Presentation:** 2
**Contribution:** 2
**Rating:** 5
**Confidence:** 4

**Summary:**

This paper introduces W2SAttack, a method for injecting clean-label backdoors into LLMs. The approach stems from the observation that successfully injecting clean-label backdoors during fine-tuning becomes challenging when using Parameter Efficient Fine-Tuning (PEFT) algorithms. The authors analyze the limitations of PEFT from the information theory perspective and propose Weak-to-Strong Attack (W2SAttack) to enhance attack effectiveness under PEFT. Inspired by teacher-student knowledge distillation, W2SAttack first injects backdoors into a smaller teacher model using full-parameter fine-tuning. It then transfers the backdoor knowledge to a larger student model through PEFT, incorporating feature alignment loss terms during the distillation process to support the backdoor learning. Evaluation results demonstrate that W2SAttack can effectively inject various types of backdoors into LLMs using PEFT.

**Strengths:**

1. The research topic is interesting and important to the community

2. The idea is novel and intuitive.

3. The paper is overall well-written and easy to follow

**Weaknesses:**

1. The threat model combining clean-label backdoor attacks with training control is counter-intuitive and lacks practical value.

2. The observation that PEFT cannot successfully inject backdoors is inconsistent with findings in recent literature.

3. The paper lacks evaluation on larger LLMs to demonstrate the scalability and effectiveness of the proposed method.

4. The paper lacks evaluation on generative tasks, which are a major use case for LLMs today.

**Questions:**

1. The paper addresses injecting clean-label backdoors into LLMs under the assumption that the attacker has full control over the training process, making this setup counterintuitive and somewhat confusing. The main advantage of clean-label backdoor attacks is their stealthiness, as they can bypass human inspection. Most existing clean-label backdoor attacks operate under a data poisoning assumption, where the attacker only provides poisoned data without controlling the training process [1, 2, 3, 4]. In this scenario, model trainers may inspect the received data before using it for training. Due to the label consistency in clean-label backdoor attacks, simple human inspection cannot detect the poisoned samples, which makes them stealthy. However, in a training control setup [5, 6], the stealth advantage of a clean-label backdoor is irrelevant because the attacker will only release the poisoned model, without exposing the poisoned training data. This means there is no data inspector in such a scenario, and attackers can freely manipulate data to ensure successful backdoor injection while maintaining benign performance. Therefore, the motivation for studying clean-label backdoors in a training control setup is unclear.

2. The paper claims that PEFT algorithms struggle to successfully inject backdoors into LLMs. According to Table I, even dirty-label attacks (e.g., BadNets) using PEFT only achieve a 15.51% ASR on the SST-2 dataset. This observation contradicts recent literature on LLM backdoors [7, 8, 9]. For example, [7] reports successful backdoor injection into LLMs using QLoRA, and [8] proposes a fine-tuning method similar to PEFT that achieves effective backdoor injection. Can the authors clarify the reasons behind these contradictory findings?

3. One of the main advantages of W2SAttack is its ability to inject backdoors into models that cannot be trained using full-parameter fine-tuning due to computational constraints. Therefore, it would strengthen the paper if the authors included results from applying W2SAttack to larger open-source LLMs, such as Llama-2-70B or Mixtral-8x7B. This would further support the argument for the proposed attack.

4. Another point of concern is that the paper focuses primarily on LLM discriminative tasks, such as sentiment classification, whereas LLMs are now predominantly used for generative tasks. Recent works have also explored backdoors in LLMs for generative tasks [10, 11]. It would be valuable if the authors extended their proposed attack to generative tasks to determine if the same observations hold in those contexts.

---

Reference

[1] Liu, Yunfei, et al. "Reflection backdoor: A natural backdoor attack on deep neural networks." Computer Vision–ECCV 2020: 16th European Conference, Glasgow, UK, August 23–28, 2020, Proceedings, Part X 16. Springer International Publishing, 2020.

[2] Barni, Mauro, Kassem Kallas, and Benedetta Tondi. "A new backdoor attack in cnns by training set corruption without label poisoning." 2019 IEEE International Conference on Image Processing (ICIP). IEEE, 2019.

[3] Zeng, Yi, et al. "Narcissus: A practical clean-label backdoor attack with limited information." Proceedings of the 2023 ACM SIGSAC Conference on Computer and Communications Security. 2023.

[4] Turner, Alexander, Dimitris Tsipras, and Aleksander Madry. "Clean-label backdoor attacks." (2018).

[5] Cheng, Siyuan, et al. "Deep feature space trojan attack of neural networks by controlled detoxification." Proceedings of the AAAI Conference on Artificial Intelligence. Vol. 35. No. 2. 2021.

[6] Doan, Khoa, et al. "Lira: Learnable, imperceptible and robust backdoor attacks." Proceedings of the IEEE/CVF international conference on computer vision. 2021.

[7] Huang, Hai, et al. "Composite backdoor attacks against large language models." arXiv preprint arXiv:2310.07676 (2023).

[8] Li, Yanzhou, et al. "Badedit: Backdooring large language models by model editing." arXiv preprint arXiv:2403.13355(2024).

[9] Li, Yige, et al. "Backdoorllm: A comprehensive benchmark for backdoor attacks on large language models." arXiv preprint arXiv:2408.12798 (2024).

[10] Rando, Javier, and Florian Tramèr. "Universal jailbreak backdoors from poisoned human feedback." arXiv preprint arXiv:2311.14455 (2023).

[11] Hubinger, Evan, et al. "Sleeper agents: Training deceptive llms that persist through safety training." arXiv preprint arXiv:2401.05566 (2024).

---

> ### Author Response · Authors · 2024-11-16
> **Response to Reviewer LsBW**
>
> Dear Reviewer LsBW,
>
> **Thank you for your review!** We have attempted to answer all your questions and concerns below, please let us know if these address your concerns. **If you feel that your concerns have been satisfactorily addressed, we would be grateful if you would consider revising your score.** Please do not hesitate to reach out with any further questions. We value your feedback and welcome any additional queries.
>
> ***
>
> **Question 1:** The paper addresses injecting clean-label backdoors into LLMs under the assumption that the attacker has full control over the training process, making this setup counterintuitive and somewhat confusing. The main advantage of clean-label backdoor attacks is their stealthiness, as they can bypass human inspection. Most existing clean-label backdoor attacks operate under a data poisoning assumption, where the attacker only provides poisoned data without controlling the training process [1,2,3,4]. In this scenario, model trainers may inspect the received data before using it for training. Due to the label consistency in clean-label backdoor attacks, simple human inspection cannot detect the poisoned samples, which makes them stealthy. However, in a training control setup [5,6], the stealth advantage of a clean-label backdoor is irrelevant because the attacker will only release the poisoned model, without exposing the poisoned training data. This means there is no data inspector in such a scenario, and attackers can freely manipulate data to ensure successful backdoor injection while maintaining benign performance. Therefore, the motivation for studying clean-label backdoors in a training control setup is unclear.
>
> **Response 1:** Thank you for your comments. The reason we chose clean-label backdoor attacks in this paper is as follows:
>
> >**Motivation:** In our pilot study, we found that under the PEFT setting, clean-label backdoor attacks struggle to achieve feasible outcomes compared to full-parameter fine-tuning. To address this issue, this paper introduces a novel backdoor attack algorithm based on feature alignment-enhanced knowledge distillation, aimed at improving the success rate of clean-label backdoor attacks under the PEFT setting. **In summary, our motivation is to enhance the effectiveness of clean-label backdoor attacks**; thus, we need to manipulate the training process.
>
> >**Effectiveness:** In real-world application scenarios, even though attackers do not need to worry about the inspection of training data when they control the training process, we still hope that clean-label backdoor attacks are more effective. Similarly, **previous studies have also manipulated the training process under the clean-label settings**. For example, Huynh et al. implement clean-label backdoor attacks using alternated training [1]; Cheng et al. implement clean-label backdoor attacks by injecting the trojan into the feature space during the model training process [2].
>
> Additionally, **existing research indicates that leveraging small-scale language models as guides has the potential to enhance the performance of LLMs. However, if this strategy is employed by attackers, it may transmit backdoor features to the LLMs, posing potential security risks. Therefore, the potential applications of W2SAttack may be utilized in weak-to-strong model scenarios, which involve poisoning LLMs in a clean-label setting**.
>
> ***
>
> **Question 2:** The paper claims that PEFT algorithms struggle to successfully inject backdoors into LLMs. According to Table I, even dirty-label attacks (e.g., BadNets) using PEFT only achieve a 15.51% ASR on the SST-2 dataset. This observation contradicts recent literature on LLM backdoors [7, 8, 9]. For example, [7] reports successful backdoor injection into LLMs using QLoRA, and [8] proposes a fine-tuning method similar to PEFT that achieves effective backdoor injection. Can the authors clarify the reasons behind these contradictory findings?
>
> **Response 2:** Thank you for your comments. Firstly, we consistently employ clean-label backdoor attacks in Table 1 of the manuscript, including the BadNets algorithm, which specifically implants triggers into poisoned samples without altering their labels.
>
> Secondly, **in references [7,8,9], attackers employed poison-label backdoor attacks, which differ from the clean-label backdoor attacks we explore**. Previous backdoor attack research achieved high ASR because poison-label backdoor attacks require modifying the labels of poisoned samples. Consequently, there is an explicit mapping relationship between the trigger and the target label, making this attack paradigm easier to learn, even with only a small number of model parameters updated.
>
> Finally, in our pilot study, we found that under the PEFT setting, clean-label backdoor attacks are challenging to carry out effectively. This differs from the poison-label backdoor attacks described in references [7,8,9].

---

> ### Author Response · Authors · 2024-11-16
> **Response to Reviewer LsBW**
>
> **Question 3:** One of the main advantages of W2SAttack is its ability to inject backdoors into models that cannot be trained using full-parameter fine-tuning due to computational constraints. Therefore, it would strengthen the paper if the authors included results from applying W2SAttack to larger open-source LLMs, such as Llama-2-70B or Mixtral-8x7B. This would further support the argument for the proposed attack.
>
> **Response 3:** Thank you for your suggestions. To validate the effectiveness of the W2SAttack algorithm, we deployed multiple state-of-the-art large language models, such as Llama-3-8B, Vicuna-7B, and Mistral-7B, conducting a total of 72 experiments. The results of all experiments consistently demonstrated that the W2SAttack significantly enhances the success rate of clean-label backdoor attacks in PEFT settings.
>
> We greatly appreciate your suggestions and understand that testing larger-scale language models would be beneficial for validating the effectiveness of the W2SAttack algorithm. However, due to hardware limitations, we are unable to include models such as Llama-2-70B or Mistral-8x7B. To address your concerns, we have conducted additional experiments on the Llama-13B model, hoping for your understanding. The experimental results, as shown in Table 1, indicate that the W2SAttack also helps to improve the ASR in the Llama-13B model.
>
> |       |SST-2|            | CR     |       | AG’s News |    |
> |:---------:|:-------------:|:---------:|:----------:|:---------:|:----------:|:---------:|
> |Method| CA | ASR | CA | ASR | CA | ASR |
> |LoRA     | 96.60|30.36 | 93.16| 16.84 | 91.24 | 27.56 |
> |W2SAttack | 95.55|99.45 | 90.58| 97.71 | 91.79 | 97.39 |
> |
>
> Table 1: The results of our W2SAttack algorithm in PEFT. The language model is Llama-13B.
>
> ***
>
> **Question 4:** Another point of concern is that the paper focuses primarily on LLM discriminative tasks, such as sentiment classification, whereas LLMs are now predominantly used for generative tasks. Recent works have also explored backdoors in LLMs for generative tasks [10,11]. It would be valuable if the authors extended their proposed attack to generative tasks to determine if the same observations hold in those contexts.
>
> **Response 4:** Thank you for your comments. Given the diversity of sample labels in generative tasks, backdoor attack research targeting these tasks often requires the modification of the labels of poisoned samples and the fine-tuning of the victim model to align triggers with target labels. Consequently, studies on backdoor attacks for generative tasks predominantly focus on poison-label backdoor attacks [3,4].
>
> The motivation of this paper is to enhance the effectiveness of clean-label backdoor attacks under the PEFT setting. Therefore, the W2SAttack is not applicable to generative tasks. We greatly appreciate your valuable suggestions and have discussed the potential differences between our study and related research on generative tasks [10,11] in the manuscript.
>
> ***
>
> **References:**
>
> [1] Huynh, Tran, et al. "COMBAT: Alternated Training for Effective Clean-Label Backdoor Attacks." Proceedings of the AAAI Conference on Artificial Intelligence. Vol. 38. No. 3. 2024.
>
> [2] Cheng, Siyuan, et al. "Deep feature space trojan attack of neural networks by controlled detoxification." Proceedings of the AAAI Conference on Artificial Intelligence. Vol. 35. No. 2. 2021.
>
> [3] Li, Yige, et al. "Backdoorllm: A comprehensive benchmark for backdoor attacks on large language models." arXiv preprint arXiv:2408.12798 (2024).
>
> [4] Yang, Zhou, et al. "Stealthy backdoor attack for code models." IEEE Transactions on Software Engineering (2024).
>
> …
>
> [7] Huang, Hai, et al. "Composite Backdoor Attacks Against Large Language Models." Findings of the Association for Computational Linguistics: NAACL 2024. 2024.
>
> [8] Li, Yanzhou, et al. "BadEdit: Backdooring Large Language Models by Model Editing." The Twelfth International Conference on Learning Representations.
>
> [9] Li, Yige, et al. "Backdoorllm: A comprehensive benchmark for backdoor attacks on large language models." arXiv preprint arXiv:2408.12798 (2024).
>
> [10] Rando, Javier, and Florian Tramèr. "Universal Jailbreak Backdoors from Poisoned Human Feedback." The Twelfth International Conference on Learning Representations.
>
> [11] Hubinger, Evan, et al. "Sleeper agents: Training deceptive llms that persist through safety training." arXiv preprint arXiv:2401.05566 (2024).
>
> ***
>
> >**In the end, thanks a lot for your detailed comments and thank you for helping us improve our work! We appreciate your thoughts on our work and we would be more than happy to discuss more during the rebuttal. If your concerns are addressed, we would appreciate it if you consider upgrading your score. Please let us know if you have any further questions. We are actively available until the end of this rebuttal period.**

---

> ### Author Response · Authors · 2024-11-19
> **Request for Discussion**
>
> Dear Reviewer LsBW:
>
> Kindly note that the author-reviewer discussion period is currently ongoing. We would greatly appreciate it if you could review our response when convenient. We earnestly request that you reconsider our manuscript and consider upgrading your score.
>
> Regards,
>
> Authors

---

> > ### Comment · Reviewer_LsBW · 2024-11-25
> > **Acknowledgement to author rebuttal**
> >
> > Dear Authors,
> >
> > Thank you for your efforts during the rebuttal process. Most of my concerns regarding the technical aspects have been addressed. However, I remain unconvinced by the motivation behind training control clean-label backdoor attacks. The primary purpose of a clean-label attack is to evade human inspection, but this becomes meaningless if there is no human inspector involved in the first place.
> >
> > In other words, under the training control threat model, where the poisoned data is never exposed to human inspection, why should we prioritize clean-label attacks over dirty-label attacks? Dirty-label backdoors can be injected into large language models (LLMs) with relative ease, making them a more straightforward option in this context.
> >
> > Therefore, I will maintain my score.

---

> ### Author Response · Authors · 2024-11-26
> **Further Response to Reviewer LsBW**
>
> Dear Reviewer LsBW,
>
> **Thank you for your review!**
>
> ***
>
> **Question 1:** In other words, under the training control threat model, where the poisoned data is never exposed to human inspection, why should we prioritize clean-label attacks over dirty-label attacks? Dirty-label backdoors can be injected into large language models (LLMs) with relative ease, making them a more straightforward option in this context.
>
> **Response 1:** Thank you for your comments. Firstly, in our response, we did not state that poisoned data would not be subject to human review. **Instead, in previous backdoor attack research [1], it is assumed that attackers control the training process under the clean-label setting**.
>
> Secondly, research has shown that using small models to guide LLMs can enhance performance in downstream tasks, and several weak-to-strong algorithms have been proposed [2,3,4]. Therefore, we believe **a potential application scenario for our W2SAttack algorithm is during the process where attackers poison the training dataset based on clean-label techniques while small-scale models are used to facilitate LLM learning**. We consider this a potential vulnerability.
>
> We have supplemented our manuscript with an explanation of the attack scenarios. On page 7 of the manuscript:
>
> > The potential applications of W2SAttack may be utilized in weak-to-strong model scenarios [2,3,4], which leverage small-scale models to enhance the performance of LLMs.
>
> On page 21 of the manuscript:
>
> > Existing research indicates that leveraging small-scale language models as guides has the potential to enhance the performance of LLMs. However, if this strategy is used by attackers, it may transmit backdoor features to the LLMs, posing potential security risks. Therefore, the potential applications of W2SAttack may be utilized in weak-to-strong model scenarios, which involve poisoning LLMs in the clean-label setting.
>
> **References:**
>
> [1] Cheng, Siyuan, et al. "Deep feature space trojan attack of neural networks by controlled detoxification." Proceedings of the AAAI Conference on Artificial Intelligence. Vol. 35. No. 2. 2021.
>
> [2] Zhou, Zhanhui, et al. "Weak-to-Strong Search: Align Large Language Models via Searching over Small Language Models." NeurIPS 2024.
>
> [3] Burns, Collin, et al. "Weak-to-Strong Generalization: Eliciting Strong Capabilities With Weak Supervision." Forty-first International Conference on Machine Learning.
>
> [4] Zhao, Xuandong, et al. "Weak-to-strong jailbreaking on large language models." arXiv preprint arXiv:2401.17256 (2024).

---

> > ### Comment · Reviewer_LsBW · 2024-11-26
> > **Further Response to the authors**
> >
> > Dear Authors,
> >
> > It seems there may be a misinterpretation of the definition of training control in this context.
> >
> > **A training control backdoor attack** implies that the attacker has full access to the entire training pipeline, including the (poisoned) dataset and the victim model into which they aim to inject the backdoor. Under this threat model, the attacker has more privileges, allowing them to modify various aspects of the training process, such as the loss function and hyperparameters.
> >
> > In contrast, **a data-poisoning backdoor attack** assumes that the attacker only has access to the poisoned dataset and no access to the victim's model or its training process.
> >
> > In Section 2.1 (Threat Model) of [1], it is explicitly stated that they focus on **the dataset-poisoning scheme** and **that the attacker acts as a data provider, supplying a dataset for image classification training via a commercial transaction or an open-source release.**
> >
> > Additionally, in Section 3.3 of [1], the proposed **Alternated Training** method does not assume access to the victim model. Instead, it utilizes a surrogate model to mimic the victim classifier's training process.
> >
> > Based on these definitions, [1] falls under the category of **a data-poisoning clean-label backdoor**, whereas the proposed **W2SAttack** aligns with **a training control clean-label backdoor**.
> >
> > I have a simple and straightforward question for the authors:
> >
> > **If I can control the training process and aim to inject a backdoor into the model, and I plan to release only the trained model publicly, why would I prefer a clean-label attack over a dirty-label attack?**
> >
> > ---
> > **Reference**
> >
> > [1] Huynh, Tran, et al. "COMBAT: Alternated Training for Effective Clean-Label Backdoor Attacks." Proceedings of the AAAI Conference on Artificial Intelligence. Vol. 38. No. 3. 2024.

---

> ### Author Response · Authors · 2024-11-26
> **Further Response to Reviewer LsBW**
>
> Dear Reviewer LsBW,
>
> **Thank you for your reply!**
>
> ***
>
> As you mentioned, under the same conditions, dirty-label attacks tend to be more effective than clean-label attacks. If attackers have control over the training process, they tend to prefer using dirty-label attacks. **However, clean-label attacks maintain the correctness of the poisoned samples' labels, making them easier to implement and helping to ensure the model's performance**. Moreover, in the work of Cheng et al.[1], they also manipulate the model training process under the clean-label setting. Therefore, we believe that clean-label backdoor attacks are also worth studying, even if the attackers only release the victim model.
>
> Furthermore, **we need to consider a new application scenario, the weak-to-strong model, which leverages small-scale models to guide LLMs**. This aligns with our algorithmic process. If the training dataset is poisoned and used to fine-tune language models, it is possible to transfer the backdoor from the weak model to the LLMs. In this setting, clean-label attacks offer more stealth than dirty-label attacks. Therefore, we believe it is necessary to consider this potential threat, which, although niche, is plausible.
>
> Furthermore, **the purpose of researching the backdoor attack algorithm is to identify potential security vulnerabilities in LLMs, not simply to achieve a 100% attack success rate**.
>
> ***
>
> **References:**
>
> [1] Cheng, Siyuan, et al. "Deep feature space trojan attack of neural networks by controlled detoxification." Proceedings of the AAAI Conference on Artificial Intelligence. Vol. 35. No. 2. 2021.

---

### Meta-Review · Area_Chair_XWzo · 2024-12-17

**Metareview:**

The paper proposes a new backdoor attack in the clean label setting.
As noted by reviewer LsBW, there seems to be an inconsistency in the considered threat model, where the attacker has full control over the training process (which is standard in some backdoor works), but still is restricted to clean-label attacks (which is a valid requirement if the attacker can only supply poisoned data that may be checked by humans).
As a result, the value of the paper is not entirely clear. While the attack strategy seems to work well, I recommend the authors clarify the threat model they consider and why it is realistic.

**Additional Comments On Reviewer Discussion:**

The discussion with reviewer LsBW was primarily focused on clarifying the threat model, and did not resolve the discrepancy between the very strong assumption of control of the training process, coupled with the weak assumption of clean label attacks

---

### Decision · Program_Chairs · 2025-01-22

Reject